*Nat Microbiol.* Author manuscript; available in PMC 2022 October 03.**

# Discovery of an Orally Active Benzoxaborole Prodrug Effective in the Treatment of Chagas Disease in Non-human Primates

**Angel M. Padilla[1],[†], Wei Wang[1],[†], Tsutomu Akama[2], David S. Carter[2], Eric Easom[2], Yvonne Freund[2], Jason S. Halladay[2], Yang Liu[2], Sarah A. Hamer[3], Carolyn L. Hodo[3], Gregory K. Wilkerson[4], Dylan Orr[1], Brooke White[1], Arlene George[1], Huifeng Shen[1], Yiru Jin[5], Michael Zhuo Wang[5], Susanna Tse[6], Robert T. Jacobs[2],[‡], Rick L. Tarleton[1],[‡],[*]**

[1]Center for Tropical and Emerging Global Diseases and Department of Cellular Biology, University of Georgia, Athens GA

[2]Anacor Pharmaceuticals, Inc, Palo Alto, CA

[3]College of Veterinary Medicine and Biomedical Sciences, Texas A&M University, College Station, TX

[4]Michale E. Keeling Center for Comparative Medicine and Research of The University of Texas MD Anderson Cancer Center, Bastrop TX

[5]Department of Pharmaceutical Chemistry, The University of Kansas, Lawrence, KS 66047

[6]Pharmacokinetics, Dynamics & Metabolism, Worldwide Research, Pfizer Inc

## Abstract

*Trypanosoma cruzi*, the agent of Chagas disease, likely infects 10's of millions of people, primarily in Latin America, and brings with it the morbidity and mortality of Chagas disease The options for treatment and prevention of Chagas disease are limited and underutilized. Here we describe the discovery of a series of benzoxaborole compounds with nanomolar activity against extra- and intra-cellular stages of *T. cruzi*. Leveraging both ongoing drug discovery efforts in related kinetoplastids, and the exceptional models for rapid drug screening and optimization in *T. cruzi*, we have identified the prodrug AN15368 that is activated by parasite carboxypeptidases to

[*]Corresponding author. tarleton@uga.edu.
[†]These authors contributed equally to the study
[‡]These authors contributed equally to the study

**Contributions:**
RJ and RT jointly conceived and directed the project and wrote the initial manuscript drafts. EE provided project administration and edited the manuscript. AP and WW contributed equally to manuscript and figure editing and conducted and directed the in vitro and in vivo experiments, with assistance from DO, BW, AG, YF and HS. RJ, TA, DC, YV, JH and YL were responsible for compound design and characterization. GW directed the NHP studies and SH and CH conducted PCR analysis for blood parasites in NHP. YJ and MW conducted mass spectrometric analysis of compound uptake and activation. ST assisted with pharmacokinetic studies in NHP.

**Competing interests:**
The University of Georgia Research Foundation, Inc. has a financial interest in the intellectual property (IP) detailed in the manuscript including the candidate compounds. Authors from the University of Georgia may receive personal compensation and research funding as governed by the University of Georgia IP Policy if the IP is licensed for commercial use. Authors formerly employed by Anacor Pharmaceuticals or Pfizer were paid employees and shareholders of Anacor/Pfizer and Tsutomu Akama, David S. Carter, Jason S. Halladay, Robert T. Jacobs, and Yang Liu are named inventors on issued patents. Other authors declare that they have no competing interests.

yield a compound that targets the mRNA processing pathway in *T. cruzi*. AN15368 was found to be active in vitro and in vivo against a range of genetically distinct *T. cruzi* lineages and was uniformly curative in non-human primates (NHPs) with long-term, naturally acquired infections. Treatment in NHPs also revealed no detectable acute toxicity or long-term health or reproductive impact.. Thus, AN15368 is an extensively validated and apparently safe, clinically ready candidate with promising potential for prevention and treatment of Chagas disease.

## Introduction

Chagas disease, caused by the protozoan parasite *Trypanosoma cruzi*, remains the highest impact parasitic disease in Latin America and one of the major causes of infection-induced myocarditis worldwide[1]. For more than 5 decades, two nitroheterocyclic compounds, benznidazole and nifurtimox, have been available for treatment of the infection, but are relatively rarely used due to their inconsistent efficacy and high frequency of side effects. Recent trials of potential new therapies have yielded disappointing results [2, 3].

Among the challenges for drug development in *T. cruzi* infection is the parasite's predominantly intracytoplasmic location in mammals and its ability to invade a wide variety of host cell types and tissues, although it shows a clear preference for muscle cells, including cardiac, skeletal and smooth muscle of the gut. The recent discovery of arrested "dormant" intracellular forms of *T. cruzi* that are relatively and transiently resistant to otherwise highly effective trypanocidal compounds [4] may partially explain why these therapeutics must be given for extended periods of time (60 days is common) but nevertheless still have a high failure rate.

Previous work from our laboratories have identified a novel class of boron-containing molecules, the benzoxaboroles [5, 6] as having potent activity against protozoans including *Trypanosoma brucei* [7], *Leishmania donovani* [8] and *Plasmodium falciparum* [9]. Screening of the Anacor benzoxaborole compound library against *T. cruzi* revealed several hits, but initial assessment of structure-activity relationships (SAR) suggested limited opportunity for improvement of potency and/or selectivity, particularly in those sub-classes previously found to have activity against *T. brucei* and *L. donovani*. Herein we have taken advantage of this benzoxaborole scaffold and the multiple natural host species for *T. cruzi*, to move rapidly from in vitro detection of trypanocidal activity in lead compounds into facile in vivo tests of efficacy in mice and ultimately in naturally infected non-human primates (NHPs). The result is identification of a class of benzoxaboroles that provide high rates of parasitological cure of *T. cruzi* infection. AN15368 from this class is the first, extensively validated and safe potential clinical candidate in over 50 years for the prevention/treatment of Chagas disease.

## Results

### In vitro activity and SAR

The initial lead benzoxaborole 6-carboxamide AN4169 (Fig. 1) provided 100% cure of mice infected with the *T. cruzi* Brazil strain [5], however rodent tolerability studies suggested that an insufficient therapeutic margin existed for further progression of this compound.

Profiting from a concurrent project evaluating analogues of AN4169 against *Trypanosoma congolense* [10], several compounds with submicromolar activity against *T. cruzi* in vitro and good metabolic stability in an in vitro mouse S9 liver fraction assay were identified, among these an ester of the 6-valine "transposed" carboxamide AN10443 (Fig. 1). Further manipulation of this compound, in particular inclusion of a methyl group at C(7) of the benzoxaborole ring as in AN11735, drastically increased in vitro activity against *T. cruzi* ($IC_{50} < 10$ nM) whereas substituents larger than methyl at C(7) ablated activity (Extended data 1A).

Structure activity relationship (SAR) development of the ester region in the C(7)-methyl series showed that potency was not significantly impacted by substituents on the benzyl ester, with the exception of the 4-(3-pyrrolidinylethyloxy) analog AN14502 and the 4-methylsulfonyl analog AN14561, which were significantly less potent (Supplementary Table S1). Physicochemical properties were more significantly affected, with most simple halogenated analogs being poorly soluble in aqueous media. Metabolic stability, as estimated by incubation with the mouse S9 liver fraction, was variable, and roughly tracked with lipophilicity (cLogD; Supplementary Table S1).

These observations prompted us to more substantially modify the ester region of the molecule through preparation of aliphatic and heterocyclic esters that would be expected to be less lipophilic, more water soluble and less susceptible to metabolism (Supplementary Table S2). Several interesting SARs emerged from this group of analogs: (1) esters containing basic amines (e.g. AN15143, AN15144, AN15658, AN15678, AN15129, AN15192, AN15078, AN14504 and AN15159) were less active than neutral compounds and (2) small aliphatic esters were quite potent except for the t-butyl ester (AN15134). The relationship between lipophilicity and solubility or metabolic stability continued to exist for these compounds and provided reasonably wide latitude for modulation of such properties by choice of ester substituent.

## In vivo activities

In addition to being very potent in vitro, the valine esters were also of generally good stability, including in mouse and human S9 liver fraction assays (Supplementary Table S3). Several also exhibited low clearance (< 20% hepatic blood flow) following intravenous dosing and good bioavailability following oral administration to mice, achieving good to excellent exposure (AUC > 10 μg•hr/mL) with low mg/kg doses (Supplementary Table S3). Concurrent testing of these compounds in vivo for the ability of a single oral dose to reduce an established focal infection in the footpad of mice over 3 days [11, 12] were very encouraging, with AN14353 emerging as the lead based on activity at reduced doses (Fig. 1b). Notably, AN14353 demonstrated rapid in vivo trypanocidal activity (Fig. 1c), had high in vitro potency for a range of *T. cruzi* isolates for different genetic lineages (discrete typing units, DTUs; extended data 1b) and could consistently resolve established *T. cruzi* infections at a dose of 25 mg/kg in a standard [5, 13] 40 day treatment protocol in wild-type mice (Fig. 1d) as well as infections in immunodeficient mice (Fig. 1e).

## Lead benzoxaboroles are prodrugs activated by a parasite serine carboxypeptidase

We next sought to understand the essentiality of the ester group for activity, as this functional group carries liabilities for hydrolytic and metabolic instability. An early indication of the importance of the ester function to anti-*T. cruzi* activity was evident from the already noted lack of potency of the t-butyl ester AN15134. Amide, N-methyl amide, ketone, ether and acylsulfonamide analogs of AN11735, as well the 1,2,4-oxadiazole ester bioisostere AN14562 all lacked activity (Supplementary Table S4). Furthermore, the expected carboxylic acid metabolite, AN14667 had ~1000-fold reduced activity on both intra- and extracellular amastigotes (Figs. 1a and 2a). Thus, the ester functionality is absolutely essential for anti-*T. cruzi* activity.

We hypothesized that the esters might be pro-drugs to the carboxylic acid and were cleaved within either the host cell or by a parasite enzyme. The high sensitivity of extracellular amastigotes to AN14353, but not to the carboxylic acid AN14667 (Fig. 2a) virtually eliminated the requirement for a host peptidase in pro-drug activation. Several candidate enzymes with potential ester cleavage activity in *T. cruzi* were identified, including a serine carboxypeptidase (CBP; TcCLB.508671.20) present at 2-18 copies in different *T. cruzi* strains [14] and a metallocarboxypeptidase 2 (TcCLB.504045.60). CRISPR-Cas9-driven disruption of all copies of the serine carboxypeptidases (Fig. 2b and c) but not the metallocarboxypeptidase (Supplementary Fig. S1) decreased in vitro sensitivity of *T. cruzi* amastigotes by up to 100-fold. Accumulation of the AN14353 analogue AN15368, and its conversion to the cleaved product in wild-type *T. cruzi* epimastigotes and amastigotes but not in the KO line, was documented by quantitative liquid chromatography tandem mass spectrometry (Fig. 2d), thus confirming the serine carboxypeptidase-dependent prodrug to drug conversion within *T. cruzi*.

## Mouse test of cure studies

Evaluation of dose proportionality of exposure with AN14353 and generation of the carboxylic acid metabolite AN14667 in vivo (Supplementary Fig. S2) suggested solubility-limited absorption of this compound, prompting us to attempt to further optimize aqueous solubility. Focusing on the ester region, a variety of more polar, non-basic substituents such as aliphatic and cyclic ethers as well as hydroxyvaline analogs (predicted to be less hydrophilic) of our lead compounds were evaluated for both solubility and in the *in vitro* trypanocidal assay (Supplementary Table S5). The highest in vitro-active compounds in this set were then evaluated and found to have anti-*T. cruzi* activity in the 2 day *in vivo* assay (Fig. 3a). Extensive in vitro and in vivo pharmacokinetic analysis and metabolic stability assays in mouse and human liver S9 fraction and both mouse and human plasma and in vivo pharmacokinetics studies (summarized in Supplementary Table S6) ultimately identified three additional compounds of particular interest – AN14817, AN15368, and AN16109.

Based on these aggregate data, we progressed these and related compounds into a series of "test of cure" assays in mice with a terminal immunosuppression period to reveal residual infection [5]. An initial screening using 10 mg/kg for 40 days identified several compounds for which the treated animals showed no parasite recovery by hemoculture and no parasite DNA detection in skeletal muscle using PCR (Fig. 3b). A more stringent test using only 20 days of

treatment (a treatment period for which benznidazole is only partially effective in generating cure;[5]) further distinguished the highest potency compounds from less potent ones (Fig, 3c). Based on these results, lower doses of the candidates were evaluated in a short-term (non-cure) treatment course of intraperitoneal infection with luciferase-expressing parasites (Fig. 3d and Supplementary Fig. S3) from which a dose of 2.5 mg/kg was identified and shown effective in a 40 day-treatment cure assay (Fig. 3e).

## Non-human primate test of cure study

AN14353, AN14817 and AN15368 were progressed to an array of preliminary safety pharmacology, genotoxicity and toxicology studies and all three compounds were found to exhibit little to no affinity for a broad array of mammalian enzymes, receptors and ion channels, were non-genotoxic in standard Ames and in vitro micronucleus studies, and did not demonstrate significant inhibition of representative cytochrome P450 enzymes at 10 uM. High dose 7-day toxicology studies did not distinguish the three compounds from each other, but the non-dose proportional exposure noted previously with AN14353 (Supplementary Fig. S2) was also seen with the benzylic ester AN14817, likely a consequence of solubility limited oral absorption. In contrast, the more hydrophilic analog AN15368 exhibited good dose proportional exposure in rats (Supplementary Fig. S4) and modest effects on hematology and clinical chemistry at 150 mg/kg, but none at 120 mg/kg/day or lower. Total plasma exposure ($AUC_{0-24\ h}$) in rats at 120 mg/kg/day was approximately 30,000 ng*hr/mL, with no evidence of drug accumulation between the first and seventh days of the study. Based on these observations, AN15368 was selected as a pre-clinical candidate for the treatment of Chagas disease and for evaluation in rhesus macaques (*Macca mulatta*) infected with *T. cruzi* via natural exposure in the U.S.

We chose to treat NHP for 60 days as this is the standard length of treatment employed for human infections and in previous clinical trials [2, 3] (Fig. 4a). Based on pharmacokinetic studies in NHPs (Supplementary Fig. S5) and allometric scaling [15](see Methods for additional details) a dose of 30 mg/kg was chosen for NHP to provide high possible rate of cure without compromising safety. To maximize the power of detecting an expected high rate of cure, 19 animals were enrolled in the single arm treatment study and 3 animals served as untreated controls. Pre-treatment data on the animals is shown in Sup table 8; the mean age of the animals in the treatment group was 19.4 years and the average minimal length of infection was 5.7 yrs.

All animals in the trial were positive by PCR for *T. cruzi* DNA in blood at one or more time points pre-treatment and were seropositive for anti-*T. cruzi* antibodies by standard facility screening tests and via our Luminex-based multiplex serological assay [16, 17] (Extended data 2 and 3 and Fig. 4). Parasites were also isolated from hemocultures of pre-treatment blood samples from 18 of the 19 animals in the treatment group and 2 of 3 control animals. Importantly, the isolated parasite lines showed a variability in genetic types (DTU), indicating a diversity of parasite lineages in these animals, despite the fact they acquire infection in the relatively restricted geographic footprint of the primate colony (Extended data 2). These latter characteristics are similar to those found in a previous U.S.-based study in cynomolgus macaques with naturally-acquired *T. cruzi* infection [17].

The primary endpoints of the trial were detection of parasite DNA in blood and culture of parasites from blood, for which all animals in the study were assayed a minimum of 7 times at 2 – 4.5-week intervals following the end of treatment (Fig. 4a). All AN15368-treated animals were negative by both assays at all time points (Fig. 4b and Supplementary Table S7). In contrast, 2 of the 3 untreated animals were positive by one or both hemoculture and PCR at multiple time points.

A secondary determinant of treatment efficacy was the detection of *T. cruzi* DNA by PCR in post-necropsy tissues. For this purpose, 9 of the 19 treated animals and 2 of the untreated controls were euthanized and tissues harvested. DNA was extracted and analyzed by PCR for *T. cruzi* kDNA using both individual and pooled tissue samples (Fig. 4a and b and Supplementary Table S8). For control animal C1, 30 of 68 (39.4%) of the single or pooled tissues samples from multiple tissues yielded positive PCR results. In contrast, for the 9 treated animals, tissue samples assayed singly or in pools (range 84-99 total sample sites/ animal) from heart, quadriceps, biceps, small and large intestine, esophagus, tongue, liver, spleen, abdominal fat and brain all were negative. Interestingly, we also failed to detect parasite DNA in non-treated control animal C2 by blood or tissue PCR despite it being positive by blood PCR and hemoculture, in sampling done pre-treatment.

A third measure of treatment efficacy was declining antibody levels to a set of recombinant *T. cruzi* proteins in the multiplex serological assay [16, 17]. Monitoring for decreases in anti-*T. cruzi* antibodies has been useful for assessing treatment efficacy in humans and conversion to seronegative is considered the standard for determining infection cure - but in many cases can take years post-treatment to achieve [18, 19, 20]. Nine of the remaining 10 treated macaques not terminated at the end of treatment were returned to the breeding colony and thus were available for continued periodic monitoring over >3 years (Fig. 4a). All animals showed declines in antibody levels to multiple recombinant proteins from *T. cruzi* over the 42 months after the end of treatment (Fig. 4c and Extended data 3). Importantly, blood PCR and hemoculture samples at these additional post-treatment time points also remained negative in all treated macaques (Extended data 2, Supplementary Table S7). Thus, the exhaustive examination of blood and tissues for parasite DNA and long-term monitoring of anti-*T. cruzi* antibodies conclusively establish a 100% efficacy of AN15368 in a population of time-variable, naturally infected NHPs harboring genetically diverse populations of *T. cruzi*.

Throughout the dosing period, macaques readily accepted food treats containing compound and no post-dose nausea or other interruption of normal activity was observed. No adverse events were noted in any of the 19 treated macaques during the 60 day treatment period and repeated physical examinations revealed no clinical signs that could be associated with drug administration. Per the blood-based health screening performed throughout the study, the mean values of the liver enzymes alanine aminotransferase (ALT) and alkaline phosphatase (ALP) and the levels of lymphocytes and monocytes in the blood were mildly elevated during the drug-treatment phase of the study and returned to pre-study values by the final blood draw at the end of the study (Supplementary Data S1). The 9 female animals that returned to the breeding colony have shown no abnormalities in the yearly examinations and have produced 13 healthy and *T. cruzi*-seronegative infants, in the first

two years following treatment. These latter numbers are wholly consistent with the fecundity rate of the colony in general. At necropsy, 3 of 11 euthanized animals (2 treated animals and 1 control) were identified on gross examination to have pale areas in the heart that could be consistent with myocardial damage associated with Chagas disease. Histological examination revealed inflammation in several cases but did not differentiate the treated from untreated (control) study animals and no *T. cruzi* amastigotes were detected in any tissues from the study animals (Supplementary Data S1). Thus, AN15368 is both highly effective in curing long-standing *T. cruzi* infections and presents no overt safety or reproductive health concerns in a 60 day course of treatment in NHPs.

## Target identification

During the course of this work, several benzoxaborole analogues of AN15368 with efficacy in the treatment of African trypanosomiasis in cattle were shown to target the Cleavage and Polyadenylation Specificity Factor (CPSF3), an important factor in mRNA processing [21, 22]. Similar to as reported in these studies, the overexpression of CPSF3 in *T. cruzi* (Extended data 4) also resulted in a 3-5 fold increase in resistance to AN15368 (Fig. 5) as well as cross resistance to other benzoxaborole analogues (Fig. 5B and Extended data 4), suggesting CPSF3 as at least one of the targets of AN15368 in *T. cruzi*. This conclusion is also supported by the marked and continuing reduction in parasite mRNA as early as 6 hrs following addition of AN14353 to *T. cruzi* -infected host cells, but not in benznidazole-treated cultures (Fig. 5c). Furthermore, introduction into *T. cruzi* of the Asn232 mutation to His in CPSF3, reported by Wall, et al [21] to disrupt binding of benzoxaborole compounds in *T. brucei brucei*, also conferred resistance to AN15368 (Fig. 5d). Lastly, and as noted above for AN14353 (Fig. 2), AN15368 and other related analogues all function as prodrugs and require activation by the *T. cruzi* CBP in order to efficiently kill intracellular *T. cruzi* (Fig. 5e and Extended data 4c), as is the case for the benzoxaboroles effective against African trypanosomes [23]. Thus, the highly effective benzoxaborole AN15368 is a prodrug that enters host cells and then *T. cruzi*, wherein it is cleaved by a *T. cruzi* peptidase. The product of this cleavage selectively targets CPSF3-mediated mRNA maturation in intracellular amastigotes.

Although the target of these benzoxaboroles in *T. cruzi* and in African trypanosomes appear to be the same, a number of the compounds with previously reported potent activity in vitro to *T. congolense*, had activity on extracellular amastigotes but not on *T. cruzi* amastigotes in host cells (Fig. 5e and Extended data 4d). Reasoning that this differential effect could be due to variable or limited entry into or metabolism by *T. cruzi* host cells, we screened for the activity of these compounds on extracellular amastigotes of *T. cruzi*. With one possible exception, all of these compounds have low nanomolar activity on extracellular amastigotes, indicating an additional selectivity of benzoxaborole activity on *T. cruzi* due to this parasite's intracellular lifestyle.

## Discussion

The benzoxaborale AN15368 is the first, highly effective compound for the treatment of *T. cruzi* infection discovered in >50 years, and the only compound to date shown to achieve unequivocal and apparently uniform cure of infection in NHP with long-term, naturally

acquired infections of diverse *T. cruzi* genetic types. AN15368 is orally bioactive and exhibited no overt toxicity in a 60 day course of treatment in NHP. Thus, AN15368 is a very strong candidate for ultimately progressing into human clinical trials.

Despite some successful vector control efforts, risk of infection with *T. cruzi* remains significant for human and other animals from the southern U.S. to southern South American. The currently available drugs benznidazole and nifurtimox suffer from variable efficacy and high rates of adverse events. Consequently, these drugs are not routinely used despite their relatively wide availability. The absence of highly effective treatments undermines the use of widespread and routine screening that would detect the usually asymptomatic early infection before irreversible damage is done [24] A relatively large number of potential candidates have been targeted for development, some for decades (reviewed in [25]), but those that have been progressed to human clinical trials [2, 3, 26] have performed significantly worse than currently available drugs.

The benzoxaboroles have become a rich source of development candidates for treatment of protozoal infections, with an apparent target of all being the mRNA-processing endonuclease, CPSF3 [10,21, 22, 23, 27, 28, 29, 30, 31]. Although it was initially hypothesized that AN15368 and the analogue AN11736 may not target CPSF3, based upon their limited effect on mRNA processing [22], subsequent work demonstrated that overexpression of CPSF3 in *T. b. brucei* induced resistance to killing by AN11736 [21] and we arrive at a similar conclusion with respect to the activity of AN15368 in *T. cruzi*, indicating CPSF3 as one likely target. Also, like a number of benzoxaboroles highly effective versus the African trypanosomes, AN15368 requires processing into its carboxylate form in order to achieve full potency and we show this activation is mediated by a parasite serine carboxypeptidase. However, unlike the case in the extracellular African trypanosomes, AN15368 must traffic unprocessed through both the host and the parasite plasma membranes in order to reach these activating enzymes. This requirement appears to account for the differential activity of a number of highly similar benzoxaboroles on African trypanosomes and *T. cruzi* and emphasizes the need to tailor drugs to match the specific biology of the pathogen, even when the processing/activation requirements and the target of the compounds are the same. Likewise, this outcome highlights the challenge of designing compounds with cross-species activity for genetically-related but biologically-diverse pathogens like the kinetoplastids.

The differential dosing requirements for the benzoxaboroles in *T. cruzi* infection, where 20 or more days of treatment is necessary for sterile cure, as compared to African trypanosomes – where a single dose effects cure in cattle [10] - is remarkable and further underscores the difficulties of drug discovery for *T. cruzi*. *T. cruzi* can invade diverse host cells types in tissues throughout the body, presenting a challenge for any one drug to reach effective levels in all tissues. Furthermore, *T. cruzi* amastigotes have recently been shown to assume a non-dividing, apparently low-metabolic state that provides substantial resistance to drugs [4, 32]. Fortunately, these properties do not prevent drug-induced sterile cure, but appear to make necessary an extended treatment course, as observed herein, and previously for other anti-*T. cruzi* drugs [32]. The NHP trial conducted as part of this study was initiated before knowledge of this dormancy property in *T. cruzi* and thus utilized daily dosing for 60 days, as is common for the currently used benznidazole and nifurtimox. Despite these

lengthy treatment periods, drug-induced resistance in *T. cruzi* has not been reported with respect to the benznidazole and nifurtimox nor have we observed resistance during the extended treatment courses using benzoxaboroles, despite the fact that all three drugs are produgs. Nevertheless, shortened or modified treatment regimens may be possible with the benzoxaboroles (Fig. 3c) as is the case for benznidazole [5, 32, 33], to further reduce this possibility. As a prodrug, AN15368 has the liability of potential selection of resistance via loss of the processing carboxypeptidase [23], although deletion of the CBP array in *T. cruzi* substantially reduces, but does not totally abolish, susceptibility to AN15368.

One significant advantage of drug discovery in *T. cruzi* is the very wide natural host range of the parasite, including most wild and domesticated mammals as well as mice, canines, and NHP, in addition to humans. In all these hosts, *T. cruzi* appears to behave similarly, infecting the same host cell types, being controlled (but rarely eliminated) by similar immune effector mechanisms, generating analogous pathologies, and being affected correspondingly by the same drugs. Animals in *T. cruzi* endemic areas, including the southern U.S., are at risk of acquiring *T. cruzi* infection and in some areas, this risk is severe, leading to 20-30%/yr new infection in some populations [34] as well as infections in zoo animals [35, 36]. The commonly used indoor/outdoor housing of NHPs in *T. cruzi*-endemic regions also results in naturally acquired *T. cruzi* infection in animals in these facilities, despite vector control and other preventative measures [37]. Infections in these NHPs mirror that in humans, initiating at different points in life and extending for decades in some cases, involving genetically and phenotypically diverse parasite populations, and leading to a diversity of immune responses and disease outcomes [17]. All these characteristics make these NHPs incredibly valuable resources for trialing anti-*T. cruzi* drugs prior to human clinical trials. The observed 100% cure with AN15368 in macaques harboring long-term infections with genetically diverse parasite lineages and without any apparent drug toxicity, bodes well for the potential safety and efficacy of AN15368 in humans. It is noteworthy that the only other documented treatment trials in *T. cruzi* -infected NHPs recorded a high degree of failures for posaconazole (100%) [38] and benznidazole (>60%; Tarleton, et al., in preparation), in agreement with the high failure rate of these drugs in human clinical trials [2, 3, 26]. We also validate the same methods used for monitoring treatment efficacy in humans, specifically serial blood PCR for *T. cruzi* DNA and changes in immune profiles, for use in NHPs, and reinforce these metrics with extensive tissue PCR in a subset of necropsied animals. Lastly, the ability to return infection-cured macaques to the breeding colony extends the utility of this resources, providing the potential to monitor disease development and the susceptibility to reinfection, for example, in hosts previously cured of *T. cruzi* infections.

Surprisingly, we also observed one apparent self-cure among the three untreated NHPs. Although documentation of spontaneous cures are relatively rare [39, 40, 41], they are not without precedent and may occur more frequently than currently appreciated [42] but have not been previously observed in this NHP colony.

The success of this project provides insights into some best practices for drug discovery in *T. cruzi* and perhaps related organisms. In addition to taking advantage of the potency of the oxaboroles in general as anti-infectives, and the med-chem knowledgebase in this class of compounds, this project also benefited substantially from the accessibility of infection

systems for screening and testing compounds. Specifically, the power of the mouse-based assays to quickly, easily and quantitatively assess in vitro-active compounds for in vivo activity was instrumental in rapidly identifying the compounds with the highest potential. Coupled with the availability of substantial numbers of NHPs with naturally acquired *T. cruzi* infections for pre-clinical validation, we should be able to avoid the clinical trial failures that have accompanied previous drug discovery efforts in Chagas Disease.

## Methods

### Compound Synthesis

All compounds used in this study were prepared as described in US Patent 10,882,272, granted January 5, 2021. The syntheses of representative compounds AN14353 and AN15368 is described in Sup Text 1.

### Parasites and mice

C57BL/6J mice were purchased from The Jackson Laboratory (Bar Harbor, ME), and B6.129S7-Ifngtm1Ts/J (IFN-gamma deficient) were bred in-house at the University of Georgia Animal Facility. The SKH-1 "hairless" mice backcrossed to C57BL/6 were a gift from Dr. Lisa DeLouise (University of Rochester). All the animals were maintained in the University of Georgia Animal Facility under specific pathogen–free conditions at 22°C, 50% humidity and in a 12:12 hs light:dark cycle. Male and female mice of 6 to 9 weeks of age were used. All mouse experiments were carried out in strict accordance with the Public Health Service Policy on Humane Care and Use of Laboratory Animals and Association for Assessment and Accreditation of Laboratory Animal Care accreditation guidelines. The protocol was approved by the University of Georgia Institutional Animal Care and Use Committee. *T. cruzi* tissue culture trypomastigotes of the wild-type Brazil strain, Colombiana strain coexpressing firefly luciferase and tdTomato reporter proteins [4], and CL strain expressing the fluorescent protein tdTomato [11] were maintained through passage in Vero cells (American Type Culture Collection) cultured in RPMI 1640 medium with 10% fetal bovine serum at 37°C in an atmosphere of 5% $CO_2$. Parasites genotypes were determined as previously described [17].

### In vitro amastigote growth inhibition and killing assays

The *in vitro* anti-*T. cruzi* amastiogote activity assay was performed and optimized based on the protocol described previously [11]. The change in tdTomato fluorescence intensity was determined as a measurement of growth over 72 hours of culture. For assaying drug effects on extracellular amastigotes, trypomastigotes were collected from infected Vero cell cultures and converted in acidic media as previously described [43]. Amastigotes (50,000/well) were incubated with 2-fold serial dilutions of compounds for 48 hours. The ATP production was used as an indication of growth in this case, and was measured by ATPlite Luminescence ATP Detection Assay System (PerkinElmer). Both fluorescence and luminescence were read using a BioTek Synergy Hybrid Multi-Mode reader equipped with the software Gene5 v 2.0 (BioTek). The dose-response curve was generated by linear regression analysis with GraphPad Prism v9.4.0 (GraphPad Software). IC50 was determined

as the drug concentration that was required to inhibit 50% of growth compared to that of parasites with no drug exposure.

### In vivo compound screens

**Rapid assays**—C57BL/6 mice were injected in the hind footpads with $2.5 \times 10^5$ tdTomato expressing *T. cruzi* (CL strain) and orally treated with a single dose of the compounds (50 mg/kg) at 2 dpi. Fluorescent intensity of the feet was measured at 2 dpi before compound administration and at 4 dpi in the Maestro in vivo imaging system equipped with the software Maestro 2.1.0 (CRi) as previously described[44]. The proliferation index was estimated as PI=[(T4d – T2d)/(mUnt4d-mUnt2d)]*100; where T4d and T2d are the fluorescence intensity of the feet of the treated animals at days 4 and 2 post-infection respectively; mUnt4d and mUnt2d are the average fluorescence intensity of the feet of the untreated animals at 4 and 2 dpi respectively.

**Cure assays**—Male or female C57BL/6 mice were intraperitoneally infected with $10^4$ trypomastigotes of the Brazil strain. Mouse infection was confirmed at 25-30 dpi by detection of CD8[+] T cells specific against the *T. cruzi* TSKb20 peptide in blood [45]. Compounds resuspended in 1% carboxymethyl cellulose and 0.1% Tween 80, were administered daily by gavage at the specified concentrations. In order to optimally detect persistent infection, immune responses in the mice were suppressed by intraperitoneal injection of four doses of cyclophosphamide every 2-3 days (200 mg/kg/day), beginning one week after the end of therapeutic treatment. At the end of the immunosuppression regimen, peripheral blood was checked under light microscope for parasites and cultured in LDNT media [5]. Mouse skeletal muscle samples were obtained at the end of the immunosuppression and processed for *T. cruzi*-DNA detection by qPCR as previously described [32].

**In vivo killing time assays**—C57BL/6 mice were infected with $2.5 \times 10^5$ Luciferase-expressing *T. cruzi* (Brazil strain) in the foot pads, and two days later one oral dose of AN14353 (25 mg/kg) was administered. The bioluminescent signal in the feet after i.p. injection of D-luciferin (PerkinElmer; 250mg/kg) was measured in a Lumina II IVIS imager (PerkinElmer).

**Low dose short treatment**—Hairless mice (SKH-1) were infected intraperitoneally with $5 \times 10^4$ Luciferase expressing *T. cruzi* (Colombiana strain) and orally treated from 12 to 22 dpi with 1, 2.5 or 5 mg/kg of AN16109 or AN15368. Bioluminiscence signal of the whole body was measure after D-luciferin injection in a Lumina II IVIS in vivo imager equipped with the Living Imaging 4.0 software (PerkinElmer).

### Generation of CBPs knockout

The CBPs knockout was produced using ribonucleoprotein (RNP) complexes as previously described [46]. Briefly, Brazil tdTomato strain epimastigotes were electroporated with RNP complexes containing SaCas9 and an sgRNA targeting CBP gene TcBrA4_0048170, plus a repair template containing stop codons in all three reading frames and the M13 sequence for use as a PCR anchor. The sgRNA targeted the 'GATTTACGTTGACCAGCCTGC" sequence. After 2 days of recovery, single-cell clones were derived by depositing

epimastigotes into a 96-well plate at a density of 0.5 cell/well by using a MoFlo Astrios EQ cell sorter (Beckman Coulter). DNA isolation was performed for clones, and primer pair of 5'-ACGTTGACCAGCCTGCAG-3' and 5'- TGTGTATGGGTCTGTGAG -3' was used for amplifying the wild type allele, while primer pair of M13-F: 5'-TGTAAAACGACGGCCAGT -3' and 5'- TGTGTATGGGTCTGTGAG -3' was used for amplifying the mutant allele.

### Western Blot

A total of $5 \times 10^7$ epimastigotes were harvested at 4°C and washed once with cold PBS. Pellets were suspended in RIPA buffer (150 mM NaCl, 20 mM Tris.HCl pH7.5, 1 mM EDTA, 1% SDS, 0.1% Triton X-100) with 1% Protease Inhibitor Cocktail (Thermo Scientific), and incubated 1 hour on ice. Then the suspension was sonicated (Sonics & Materials, model 501) for 10 sec using microtip probe at 25 amplitude, and the sonicate centrifuged at 16,000 $g$ for 10 min to remove the pellets and obtain total protein. Western Blot was performed according to the general established protocol. TcCBP-specific antibody (the gift of Drs. Juan José Cazzulo and and Gabriela Niemirowicz at Instituto de Investigaciones Biotecnológicas, Buenos Aires, Argentina) was diluted at 1:500 and β-tubulin antibody was diluted at 1:1000. The IRDye 800CW Donkey anti-Rabbit IgG (Li-COR) was used as secondary antibody for both TcCBP and tubulin at 1:10,000 dilution. Images were taken in the BioRad ChemiDoc imaging system with the software ImageLab Touch 2.4.03.

### Generation and confirmation of the CPSF3 overexpression line

The *CPSF3* gene was amplified using primers 'ATGCTCCCTGCGGCAGCAGCAGTAA' and 'TTACACAGCCTCCTCTGGCAAAGGCT', and integrated into the pTrex vector [47] by NEBuilder HiFi DNA assembly (New England Biolabs). The construct of pTrex-CPSF was then transfected into Brazil tdTomato epimastigotes and selected by 60ug/ml blasticidin.

To confirm the CPSF overexpression in the selected transfectants, RNA was extracted as previously described [14] and converted to cDNA using SuperScript Reverse Transcriptase (Invitrogen). Quantitative PCR reactions were performed in triplicate on the C1000 Touch Bio-Rad CFX96 real-time PCR detection system for CPSF using primer sets CPSF-1 (5'-TGAAACAGCAGCATGCCAAC-3' and 5'-CGCGTCTGTCTACCATCAGA-3') and CPSF-2 (5'-CGGCTCATTCTGATGGTAGACA-3' and 5'-TGTGCGTTGCACACTGAATG-3') in both control and CPSF over-expressing parasites, and the expression level was normalized to tubulin which was amplified using primers: 'AAGTGCGGCATCAACTACCA' and 'ACCCTCCTCCATACCCTCA'.

### Generation of CPSF3 mutant

To generate CPSF3 mutant, a RNP c omplex was transfected into Brazil tdTomato strain epimastigotes to target the *CPSF3* gene (TcBrA4_0124800), together with a repair template that contained the mutation of $Asn^{231}$ to $His^{21}$. The sgRNA targeted the 'TCTGATTGCGGAAAGCACAA' site. After 24 hours of recovery, 20 uM AN15368 was added to the parasites to select CPSF3 mutants that were resistant to drug treatment. The

ultimate resistant parasites were validated to have acquired the $Asn^{231}His$ mutation in CPSF3 via sequencing.

### RNA-seq sample preparation, sequencing and analysis

Vero cells ($10^6$) were infected with $10^7$ CL strain trypomastigotes of *T. cruzi* for 2 days before treating with either 5 uM benznidazole or 30 nM AN14353. The drug concentration used for treatment was set at 5 times the $IC_{50}$. Samples were collected at several time points for RNA extraction as previously described [48]. rRNA-depleted RNA library construction and RNA sequencing using Illumina Nextseq 75PE was carried out by Georgia Genomics and Bioinformatics Core (GGBC, University of Georgia, Georgia). Illumina reads with mean quality lower than 30 (Phred Score based) were removed from analysis, then mapped to CL Brener genome (TritrypDB release-33) and African green monkey genome [49] using the HiSAT software package v0.1.6[50] with default parameters. The mapping rate was quantified by HTseq v0.6.1 [51].

### LC-MS/MS analysis of intracellular AN15368 and AN14667

Wild-type and peptidase knockout *T. cruzi* epimastigotes ($5\times10^8$) were treated with AN15368 (10 uM) or with DMSO vehicle control for 6 hours. The cells were then pelleted and resuspended in 100 μL of PBS. The cell suspension was mixed with 200 μl of acetonitrile and centrifuged at 735 *g* for 10 minutes at room temperature. After extraction, the supernatant was further diluted with methanol:water 30:70 (v/v) containing 0.4 nM internal standard (IS) AN14817 to a concentration within the calibration range. Each sample was diluted in triplicate to provide technical replicates and the diluted sample (10 μL) was injected for subsequent LC-MS/MS analysis.

LC-MS/MS analysis was performed on a Waters ACQUITY I-Class UPLC system coupled to a Xevo TQ-S triple quadrupole mass spectrometer. An ACQUITY UPLC BEH C18 column (130 Å, 1.7 μm, 2.1 mm x 50 mm) was used for chromatographic separation, and the column temperature was 40 °C. The mobile phase consisted of water (A) and methanol (B), both containing 0.1% (v/v) formic acid. The following gradient elution was performed at a flow rate of 0.4 mL/min: 0 – 0.5 min, 30% B; 0.5 – 3 min, 30 – 95% B; 3 – 4 min, 95% B; 4 – 4.1 min, 95 – 30% B; and 4.1 – 5 min, 30% B. The MS ionization was carried out in the positive electrospray ionization (ESI) mode with following conditions: capillary voltage = 1.50 kV; desolvation temperature = 500 °C; desolvation gas flow = 1000 L/h; and nebulizer gas pressure = 7.0 bar. The MS/MS transitions used for detection and quantification were 390.1→174.9 for AN15368, 292.0→174.9 for AN14667, and 416.1→109.0 for AN14817. Data were processed using TargetLynx v4.1 software (Waters).

### NHP resource and facilities

All NHP utilized for these studies were acquired from the approximately 1000-animal, Rhesus Macaque (*Macaca mulatta*) Breeding and Research Resource housed at the AAALAC accredited, Michale E. Keeling Center for Comparative Medicine and Research (KCCMR) of The University of Texas MD Anderson Cancer Center, in Bastrop. TX. This is a closed colony, which is specific pathogen free (SPF) for Macacine herpesvirus-1 (Herpes B), Simian retroviruses (SRV-1, SRV-2, SIV, and STLV-1), and *Mycobacterium*

*tuberculosis* complex. All animals are socially housed in shaded, temperature-regulated indoor-outdoor enclosures with numerous barrels, perches, swings, and various feeding puzzles and substrates to mimic natural foraging and feeding behaviors. Standard monkey chow, *ad libitum* water, and novel food enrichment items are provided daily. Study animals that were seropositive for *T. cruzi* had acquired the infection naturally through exposure to the insect vector of the parasite while in their indoor-outdoor housing facilities. The NHP experiments were performed at the KCCMR and all protocols were approved by the MD Anderson Cancer Center's IACUC, and followed the NIH standards established by the Guide for the Care and Use of Laboratory Animals [52].

## Pharmacokinetic (PK) analysis in NHP

Pre-treatment PK analysis of AN15368 distribution and clearance was performed to assist in determining the treatment dosing regimen. While under sedation/general anesthesia AN15368 was administered at various dosing levels either intravenously (i.v.) or via oral gavage to a *T. cruzi* -seronegative rhesus macaque, and 500 uL blood samples were collected prior to dosing and at 2, 5, 15, 30 and 60 minutes post-dose administration at which time the animal was recovered from general anesthesia. Additional 500 uL blood samples were collected at 3, 6, 9, and 24-hours post-dose administration under light-anesthesia/ sedation. After the initial IV/PO PK assessment, a pre-regimen PK assessment of oral dosing was conducted with administration of a single dose of AN15368 in 3 animals over 3 dosing periods, with AN15368 (30 or 50 mg/kg dose) administer in food treats. For this pre-regimen phase, blood samples were collected at pre-dose, then at 0.25, 0.5, 1, 3, 6, 9 and 24 hours post-dose.

Mid- and end-regimen PK assessments were also performed. A "peak and trough" mid-regimen (day 30) PK assessment was performed on 3 treated animals: 1) blood was collected prior to drug dosing; 2) the animals were gavage-dosed with AN15368 in pumpkin slurry; and 3) a second blood sample was collected three hours post-dosing. The end-regimen PK analysis was performed on the 60th (and final) day of AN15368 dosing, with 18 of the 19 treated animals utilizing a non-serial, sparse sampling design. For this study 3 animals had blood collected prior-to being provided the AN15368 in food treats. The other 15 animals were provided AN15368 in food treats and then blood was collected from 3 separate animals at 0.5, 1, 3, 6 and 9 hours post-dosing. Only 1 blood sample was collected at a single time point from each of the 18 animals. The composite plasma PK profile on Day 60 was obtained using the mean concentrations (n=3) at each sampling timepoint (predose, 0.5, 1, 3, 6 and 9 hr). All blood samples (500 uL) for PK analysis (pre-, mid- and end-regimen) were collected into EDTA microtainers and plasma harvested for the determination of AN15368. The plasma samples were provided to Pharmout Labs (Fremont, California) for analysis using LC-MS/MS. The mean pre-dose concentration was also depicted as the 24 hr post-dose concentration and used for the calculation of post-treatment $AUC_{0-24}$ value on Day 60.

For calculation of PK parameters, the $C_{max}$ (maximum concentrations) and $t_{max}$ (time to maximum concentrations) were determined by visual inspection of the plasma concentration vs. time curves from the pre-regimen and end-regimen periods. PK calculation was not

performed for the mid-regimen PK samples since only 2 time points were collected. The AUC values for the pre- and end-regimen were calculated using the linear-trapezoidal rule with the following equation:

$$AUC_{(t1 - t2)} = [(C_{t2} + C_{t1}) x (t2 - t1)]/2$$

where t1 and t2 are consecutive sampling time points, $AUC_{(t1\text{-}t2)}$ is the fractional area-under-the-curve over time intervals t1 and t2, $C_{t2}$ is the concentration at time t2, $C_{t1}$ is the concentration at time t1. The total AUC ($AUC_{0\text{-}24}$) over the dosing interval (24 hr) was calculated by summation of all fractional AUC values over the intervals between 0 (pre-dose) and 24 hours post-dose.

When a terminal elimination phase was apparent in the plasma concentration vs. time curve, the terminal half-life ($t_{1/2}$) was estimated using the equation $t_{1/2} = 0.693/\lambda_z$, where $\lambda_z$ was the elimination rate constant estimated from the slope of the terminal elimination phase. $AUC0\text{-}\infty$ was estimated using the following equation:

$$AUC_{0 - \infty} = AUC_{last} + C_{last}/\lambda_z$$

Where $AUC_{0\text{-}\infty}$ was AUC from zero to infinity, $AUC_{last}$ was the AUC from zero to the last measurable time point, $C_{last}$ was the concentration at the last measurable time point. $AUC_{0\text{-}\infty}$ and $t_{1/2}$ were not estimated when the terminal elimination phase was not defined.

Plasma clearance ($CL_p$) after the IV dose was estimated using the following: $CL_p = Dose/AUC_{0\text{-}\infty}$. Bioavailability (%F) after a single oral dose was estimated using the following:

$$\%F = \left(AUC_{0 - \infty, PO}/Dose_{PO}\right)/\left(AUC_{0 - \infty, IV}/Dose_{IV}\right)$$

Mean AN15368 plasma concentrations and PK parameters after a single IV or PO dose and mean AN15368 (total and free) plasma concentrations and PK parameters in the pre- and end-regimen periods are depicted in in Fig S4.

### NHP treatment study

A total of 22 rhesus macaques that had been confirmed to be serologically- and PCR-positive for *T. cruzi* were utilized in these studies. Using 19 animals in the treatment group provided 85% power of detecting 100% efficacy. The 19 animals were treated with a 30 mg/kg dose of AN15368 delivered in food treats once a day for 60 days. The remaining three animals on the study were maintained as untreated control animals and received food treats but were not dosed with AN15368.

The selected dose of 30 mg/kg in NHP was determined based upon the following rationale. The minimal efficacious dose in mice was determined to be 2.5 mg/kg (Fig 3e) and pharmacokinetic (PK) studies in mice at a 10mg/kg dose yielded an exposure of 17.5 (AUC0-last (µg•hr/kg); Table S6). Assuming linear exposure, a minimal curative exposure in mice was estimated at 4.375 µg•hr/kg (i.e. 17.5/4). PK analysis of a 30 mg/kg dose in

NHP indicated an exposure of 3.5-4.7 μg•hr/kg (Fig S5). Based on allometric scaling[15] the NOAEL of 120 mg/kg in rats translates to a NOAEL dose of 60.5 mg/kg in monkeys and 19.4 mg/kg in humans. Thus, the 30 mg/kg NHP dose was expected to achieve an efficacious level based upon PK comparisons of mouse and NHP and be safe as it is well below the estimated NOAEL dose determined in rats.

Under light-anesthesia/sedation, peripheral blood samples were collected from each animal prior to treatment and at 7 time points following. Blood analysis (CBC and serum chemistry assays) and physical exams to evaluate the health of the study animals were performed on each animal prior to the beginning of the study and also at three time points during the treatment protocol (see Supplementary data). At the termination of the study, 9 treated and 2 control animals were euthanized, necropsied, and blood and tissues were evaluated histologically and using PCR and hemoculture for evidence of active *T. cruzi* infection. The remaining 10 treated and 1 control animals from the study were returned to the breeding colony at the Keeling Center.

### NHP blood and tissue PCR for *T. cruzi* DNA and hemoculture

Blood samples from each macaque were collected at various time points and processed for quantification of *T. cruzi* DNA by real-time qPCR. Between 8-10ml of whole blood collected in EDTA anticoagulant tubes was subjected to DNA extraction using the Omega E.Z.N.A. Blood DNA Maxi Kit (Omega Bio-Tek), following the manufacturer's instructions for up to 10 ml whole blood and using a total of 650 μl of elution buffer. Each round of extractions included a negative (no-template) control comprised of 10 ml PBS. The concentration of DNA in the eluted solution was quantified after each extraction using an Epoch microplate spectrophotometer (BioTek).

DNA from each sample was then subjected to a series of two qPCR assays for detection of *T. cruzi* satellite DNA. The first qPCR used the cruzi 1, 2 primer set and cruzi 3 TaqMan probe as previously described [53, 54], using BioRad iTaq Universal Probes Supermix (Bio-Rad). This qPCR amplifies a 166-bp region of a repetitive satellite DNA sequence and is sensitive and specific for *T. cruzi* when compared to other PCR techniques [55]. In order to rule out false negative PCR results due to inhibition, an internal amplification control (IAC) was added to the second qPCR reaction, which was run as a multiplex as previously described, with the cruzi 1/2/3 primers and probe and the IAC primers and probe [54], except that the IAC sequence was synthesized as a gene fragment by a commercial laboratory (gBlocks Gene Fragments, Integrated DNA Technologies), and was added at the time of PCR, rather than before extraction. Positive (DNA extracted from *T. cruzi* Sylvio X10 clone 4, American Type Culture Collection, ATCC #50800, known concentration $1.7 \times 10^{-3}$ parasite equivalents) and negative (water) controls were included in each PCR plate for both assays. C1000 Touch Bio-Rad CFX96 real-time PCR detection system was used for both assays under the following cycling conditions: (i) initial denaturation, 95°, 3 min; (ii) denaturation, 95°C, 15 s; (iii) annealing, 58°C, 1 min; (iv)×50 cycles. We selected FAM and VIC channels for each read.

Frozen tissues were screened for *T. cruzi* DNA using 8mm biopsies (Sklar instruments #96-1130), collecting 3 to 10 individual ~100 μl tissue punches for each tissue type and one

of more pooled sample consisting of five punches from different areas of the tissue, totaling ~500 μl per pool. The tissues sampled included liver, heart, fat, esophagus, quadricep, bicep, large intestine, and brain, as well as tongue and spleen in a few instances. Two individual and one pooled sample of tissues from an uninfected macaque was collected for each sampling batch. DNA from macaque tissue was extracted and analyzed as previously described [32] with the exceptions that the purification was scaled up to accommodate the larger amount of tissue in the pooled samples and the range for the standards was $2.6 \times 10^2$ – $2.6 \times 10^{-3}$ parasite equivalents using the kDNA minicircle S35 and S36 primers. The Biorad CFX manager software version 3.1 was used to analyze PCR data. For samples to be considered positive, both replicates per sample must show a product (cQ value) of <40 and less than that of the included naïve sample and melt curves had to be in the same temperature range as the standards for each plate.

For hemoculture determinations, peripheral blood from macaques was collected and cultured at 26°C in supplemented liver digest neutralized tryptose medium as described previously [17]. The presence of *T. cruzi* parasites was assessed every week for 3 months under an inverted microscope. *T. cruzi* DTU of the macaque isolates was determined as previously described [17].

### Multiplex serological analysis

Luminex-based multiplex serological assays were performed as previously described [16, 17]. For a number of smaller proteins, fusions of up to 2 individual genes are employed for some target proteins in order to expand the array of antibodies being detected while controlling costs and complexity of the assay. (TritrypDb.org identifiers: Tc1= fusion of TcBrA4_0116860 and TcYC6_0028190; Tc2= fusion of TcBrA4_0088420 and TcBrA4_0101960; Tc3 = fusion of TcBrA4_0104680 and TcBrA4_0101980; Tc4= fusion of TcBrA4_0028480 and TcBrA4_0088260; Tc5 = fusion of TcYC6_0100010 and TcBrA4_0074300; Tc7 = fusion of TcYC6_0083710 and TcBrA4_0130080; Tc8 = TcYC6_0037170; Tc11 = TcYC6_0124160; Tc17 = fusion of TcBrA4_0028230 and TcBrA4_0029760; Kn107 = TcCLB.508355.250; G10 = TcCLB.504199.20). Macaque antibody binding to individual beads in the Multiplex assays was detected with donkey anti-human IgG (H+L) conjugated to phycoerythrin (Jackson ImmunoResearch) in a 1:200 dilution.

### Statistics and Reproducibility

The non-parametric Mann-Whitney U test and the unpaired t-test from the software GraphPad Prism v9.4.0 were used. Values are expressed as mean ± SEM. Statistical significance of P values was considered as *=p 0.05; **=p 0.01; ***=p 0.001. All mouse experiments were performed at least twice with similar results. All in vitro parasite proliferation assays were repeated at least one time with similar results. PCR and Western blot assays depicted as representative microphotographs in Fig 2b were repeated three and one time respectively with similar results. The quantitative liquid chromatography tandem mass spectrometry assay described in Fig. 2d was performed once. Due to cost and complexity, the NHP trial was performed once..

## 1 Extended Data

**a**

AN10443
*T. cruzi* IC$_{50}$ = 670 nM

AN11622
*T. cruzi* IC$_{50}$ > 1250 nM

AN11735
*T. cruzi* IC$_{50}$ = 4 nM

AN14557
*T. cruzi* IC$_{50}$ = 37 nM

AN14577
*T. cruzi* IC$_{50}$ > 1250 nM

AN14630
*T. cruzi* IC$_{50}$ > 1250 nM

**b**

| Strains | DTU | AN14353 | |
|---|---|---|---|
| | | IC$_{50}$ (nM) | IC$_{90}$ (nM) |
| Colombiana | TcI | 7 | 10 |
| Montalvania | TcI | 1 | 2 |
| 20290 | TcI | 3 | 5 |
| ARC0704 | TcI | 3 | 7 |
| Tul8 | TcII | 1 | 2 |
| M5631 | TcIII | 1 | 3 |
| 20392 | TcIV | 5 | 10 |
| CL | TcVI | 6 | 18 |

**Extended Data Fig. 1. Structure activity relationships of benzoxaboroles and their activity on different genetic types of *T. cruzi*.**
**a**. Substitutions in the benzoxaborole ring C(7) position of AN10443 strongly affects in vitro activity against *T. cruzi*, with larger substitutions decreasing activity. **b**. *T. cruzi* of diverse genetic types (DTUs) are susceptible to AN14353-mediated killing (n=3 biological replicates for each profile determination)

| Treatment | Male | Age (yr) | Year seropositive | Years seropositive at study start | Pre-treat blood PCR | Pre-treat hemoculture | T. cruzi genotype | days post treatment ( blood PCR/hemoculture ) | | | | | | | | | | |
|---|---|---|---|---|---|---|---|---|---|---|---|---|---|---|---|---|---|---|
| | | | | | | | | 0 | 20 | 34 | 54 | 68 | 103 | 131 | 145 | 460 | 945 | 1281 |
| T1 | . | 20 | 2012 | 6 | + | + | TCI | -/- | -/- | -/- | -/- | -/- | -/- | -/- | euth/Hem- | | | |
| T2 | . | 20 | 2014 | 4 | + | + | TCI | -/- | -/- | -/- | -/- | -/- | -/- | -/- | euth/Hem- | | | |
| T3 | . | 21 | 2013 | 5 | + | + | TCIV | -/- | -/- | -/- | -/- | -/- | -/- | -/- | euth/Hem- | | | |
| T4 | 1 | 19 | 2011 | 7 | + | + | TCI | -/- | -/- | -/- | -/- | -/- | -/- | -/- | euth/Hem- | | | |
| T5 | . | 19 | 2010 | 8 | + | + | TCI | -/- | -/- | -/- | -/- | -/- | -/- | -/- | euth/Hem- | | | |
| T6 | . | 16 | 2014 | 4 | + | + | TCI | -/- | -/- | -/- | -/- | -/- | -/- | -/- | euth/Hem- | | | |
| T7 | . | 22 | 2013 | 5 | + | + | TCI | -/- | -/- | -/- | -/- | -/- | -/- | -/- | euth/Hem- | | | |
| T8 | . | 19 | 2013 | 5 | + | - | | -/- | -/- | -/- | -/- | -/- | -/- | -/- | euth/Hem- | | | |
| T9 | . | 19 | 2008 | 11 | + | + | TCIV | -/- | -/- | -/- | -/- | -/- | -/- | -/- | euth/Hem- | | | |
| T10 | . | 5 | 2016 | 2 | + | + | TCIV | -/- | -/- | -/- | -/- | -/- | -/- | -/- | | -/- | -/- | -/- |
| T11 | . | 12 | 2014 | 4 | + | + | TCIV | -/- | -/- | -/- | -/- | -/- | -/- | -/- | | -/- | -/- | -/- |
| T12 | . | 11 | 2015 | 3 | + | + | TCI | -/- | -/- | -/- | -/- | -/- | -/- | -/- | | -/- | -/- | -/- |
| T13 | . | 11 | 2015 | 3 | + | + | TCI | -/- | -/- | -/- | -/- | -/- | -/- | -/- | | -/- | -/- | -/- |
| T14 | . | 9 | 2014 | 4 | + | + | TCI | -/- | -/- | -/- | -/- | -/- | -/- | -/- | | -/- | -/- | -/- |
| T15 | . | 16 | 2011 | 7 | + | + | TCIV | -/- | -/- | -/- | -/- | -/- | -/- | -/- | | -/- | -/- | -/- |
| T16 | . | 20 | 2010 | 8 | + | + | TCIV | -/- | -/- | -/- | -/- | -/- | -/- | -/- | | -/- | -/- | -/- |
| T17 | . | 21 | 2010 | 8 | + | + | TCIV | -/- | -/- | -/- | -/- | -/- | -/- | -/- | LTF | | | |
| T18 | . | 22 | 2010 | 8 | + | + | TCI | -/- | -/- | -/- | -/- | -/- | -/- | -/- | | -/- | -/- | -/- |
| T19 | . | 23 | 2012 | 6 | + | + | TCI | -/- | -/- | -/- | -/- | -/- | -/- | -/- | | -/- | -/- | -/- |
| **Totals/Means** | **1** | **19.4** | | **5.7** | | **19** | | | | | | | | | | | | |
| | | | | | | | | | | | | | | | | | | |
| untreated controls | Male | Age (yr) | Year positive | | | | | | | | | | | | | | | |
| C1 | . | 3 | 2017 | | + | + | TCIV | -/- | +/+ | +/+ | -/+ | +/+ | +/- | -/- | euth/Hem+ | | | |
| C2 | . | 22 | 2012 | | + | + | TCIV | -/- | -/- | -/- | -/- | -/- | -/- | -/- | euth/Hem- | | | |
| C3 | 1 | 5 | 2016 | | + | - | | +/- | +/- | +/- | +/- | -/- | +/- | +/+ | | | | |
| **Totals/Means** | **1** | **5** | | | **3** | **3** | | | | | | | | | | | | |

**Extended Data Fig. 2. Pre-, and post-treatment infection status of NHP.**
Euth/Hem = hemoculture results of blood collected at time of euthanasia. LTF = lost to follow-up due to the animal's death from traumatic lesions it acquired through conspecific interactions.

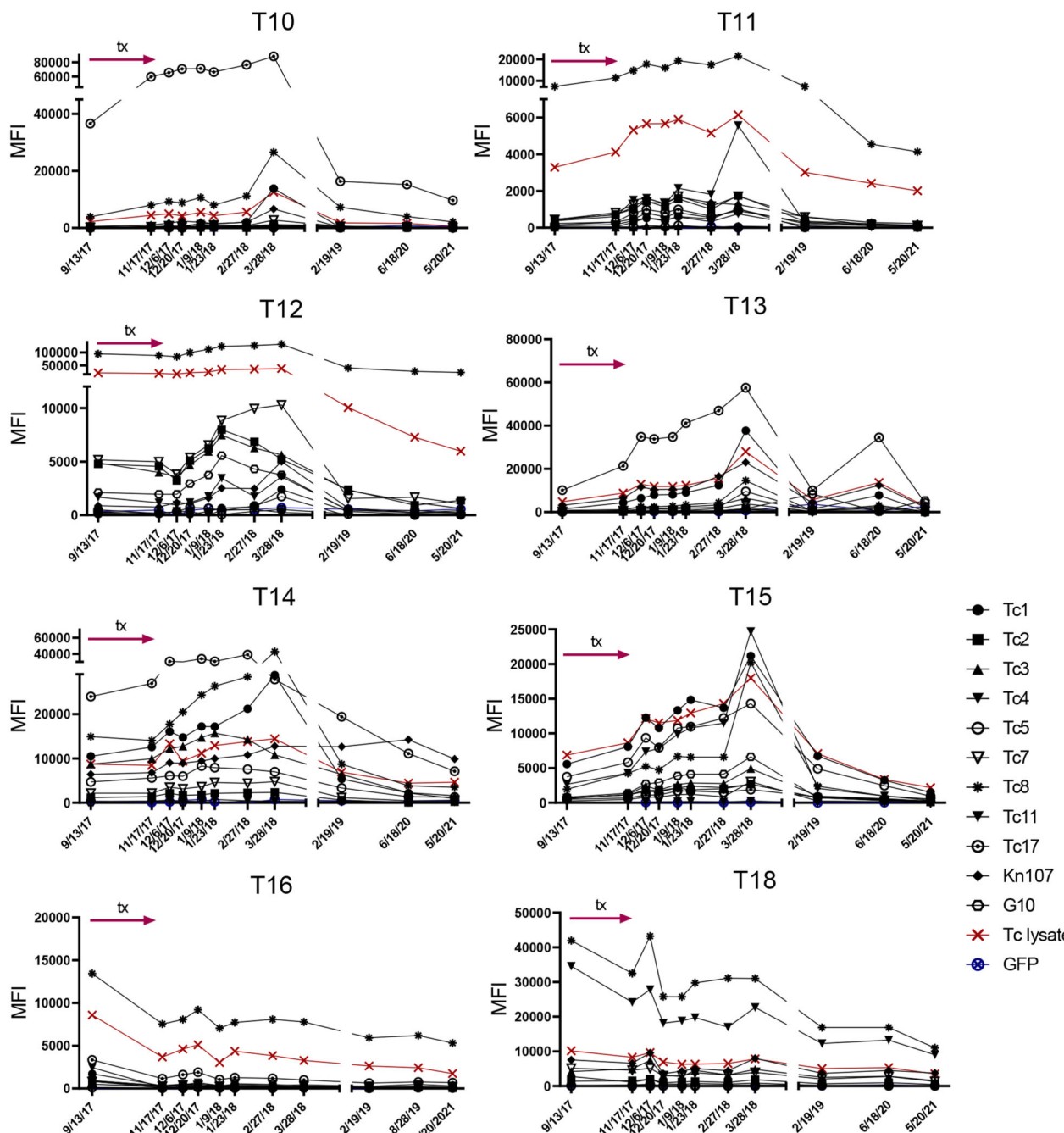

**Extended Data Fig. 3. Declining IgG levels to recombinant *T. cruzi* proteins over time in AN15368-treated NHP.**

Luminex-based pre- and post-treatment IgG responses to recombinant *T. cruzi* proteins in AN15368-treated macaques not pictured in Fig. 4 (see Methods for identification of recombinant proteins).

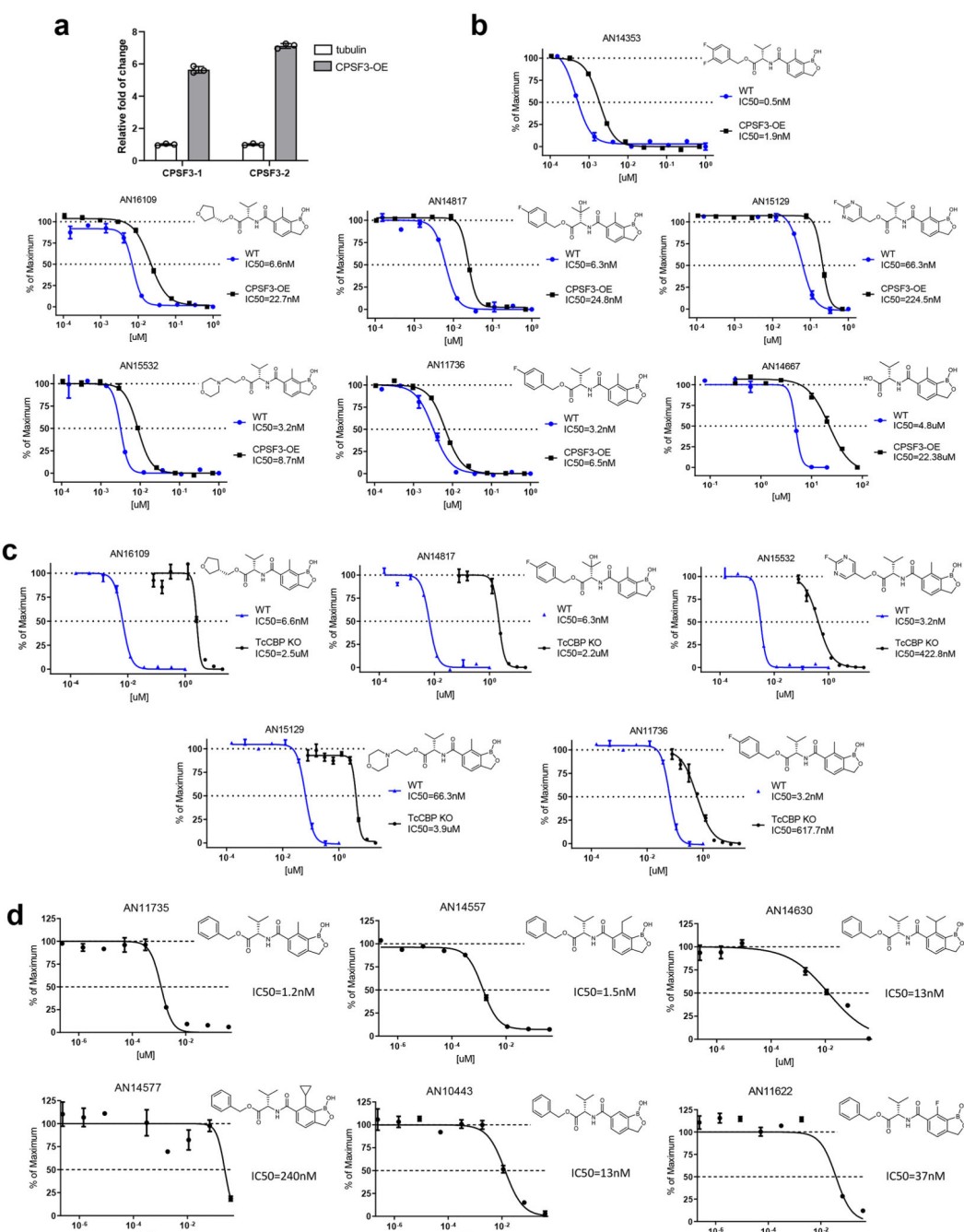

**Extended Data Fig. 4. AN15368 is activated by a *T. cruzi* serine carboxypeptidase and targets CPSF3.**

**a**. Fold change in CPSF3 transcripts by qRT-PCR in two CPSF3-overexpressing *T. cruzi* lines (n=3 biological replicates). **b**. Overexpression of CPSF3 in *T. cruzi* results in increased resistance to all AN15368 analogues (CSPF3-OE: n=3 biological replicates; WT: n=2 biological replicates). **c**. AN15368 analogues require activation by the *T. cruzi* CBP in order to efficiently kill intracellular *T. cruzi* as demonstrated by the increased resistance of CBP deficient parasites (TcCBP KO: n=3 biological replicates; WT: n=2 biological replicates).

**d**. AN15368 analogues with poor activity against intracellular *T. cruzi* amastigotes have low nanomolar activity on extracellular amastigotes (n=3 biological replicates). Data are presented as mean values +/- SD.

## Supplementary Material

Refer to Web version on PubMed Central for supplementary material.

## Acknowledgments

The authors thank Dr. Duo Peng for assistance with RNAseq analysis, Julie Nelson and the Cytometry Shared Resource Laboratory in the Center for Tropical and Emerging Global Diseases at UGA for multiplex serological analysis and flow sorting, and Drs. Juan José Cazzulo and Gabriela Niemirowicz for the gift of anti-CBP antibodies Support from the Wellcome Trust (grant number 04059/Z/14/A) to RJ and RT is acknowledged.

## Data availability

With the exception of mRNASeq data (present in the NCBI Sequence Read Archive (SRA; http://www.ncbi.nlm.nih.gov/sra/) under accession numbers are: SRX13363525 - SRX13363532), all data are available in the main text or the supplementary materials. CL Brener genome (TritrypDB release-33); (https://tritrypdb.org/tritrypdb/app). African green monkey genome (The DNA Data Bank of Japan (DDBJ, http://www.ddbj.nig.ac.jp); accession number: DRA002256)

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

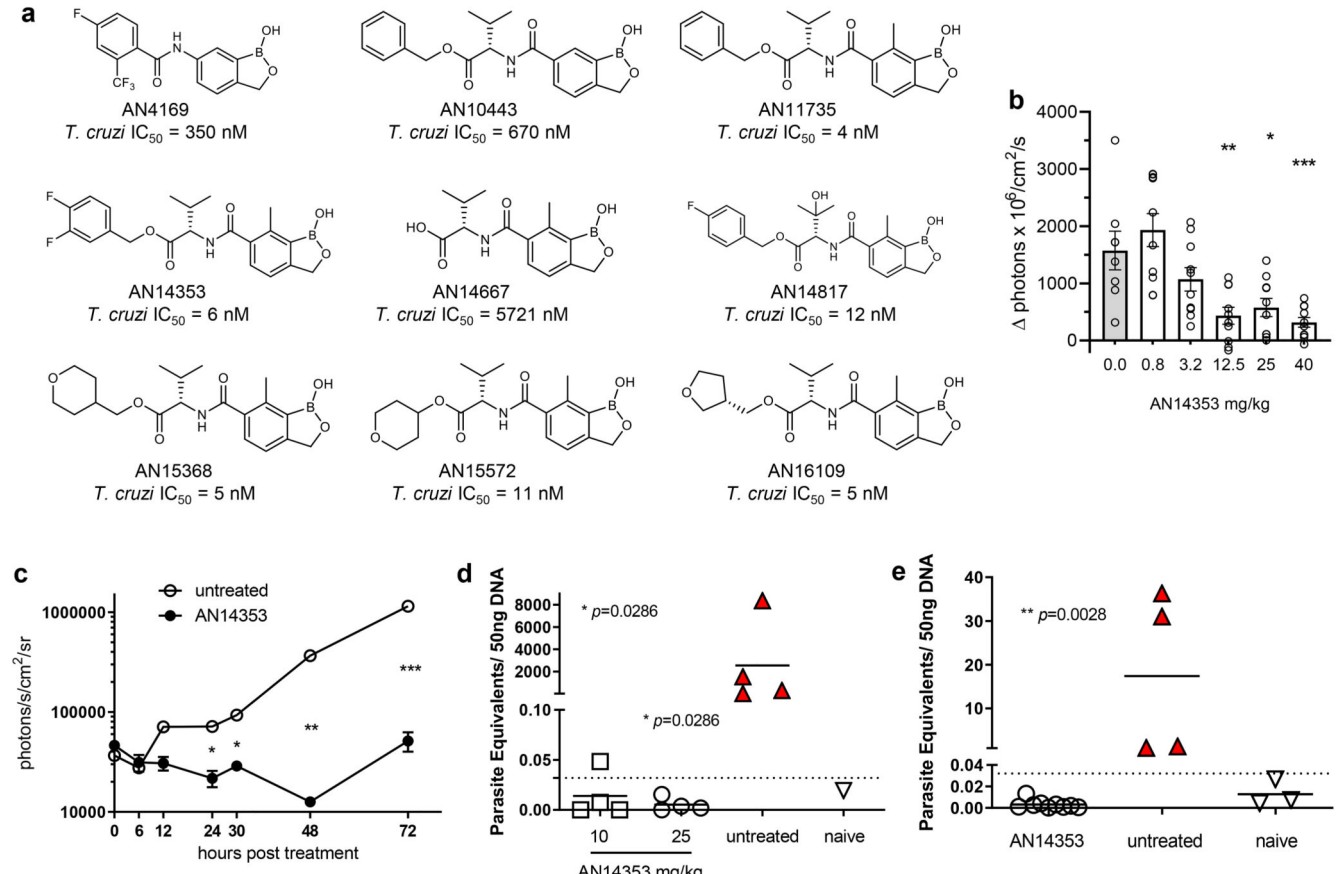

**Fig. 1. Selected benzoxaboroles are active against *T. cruzi* in vitro and in vivo.**
**(a)** Structure of key compounds evaluated, including the ultimate clinical candidate,
AN15368. **(b)** C57BL/6 mice received a single oral dose of AN14353 at the indicated
concentration at 2 dpi with $2.5 \times 10^5$ tdTomato-expressing *T. cruzi* infection in the footpads.
The relative increase in the fluorescent intensity of the footpads between 2 and 4 dpi was
assessed by in vivo imaging (untreated: n=8; AN15368: n=10 individual feet). Asterisks
represent the statistical significance of the difference between each treated group and
the untreated control. **(c)** C57BL/6 mice infected in the footpads with $2.5 \times 10^5$ Luciferase-
expressing parasites received a single oral dose of AN14353 (25 mg/kg) or no treatment
at 3 dpi and the bioluminescence signal of the feet was measured over 72 hrs (untreated:
n=2; AN15368: n=6 individual feet). **(d)** Daily oral treatment with AN14353 at 10 or 25
mg/kg was administered for 40 consecutive days to C57BL/6 mice starting at 15 days
post-intraperitoneal infection with $10^4$ *T. cruzi* trypomastigotes. Parasite load in skeletal
muscle was determined by qPCR following cyclophosphamide-induced immunosuppression
(untreated and treated groups: n=4 mice; naïve n=1). **(e)** IFN-gamma deficient mice
intraperitoneally infected with *T. cruzi* received a daily oral treatment with AN14353 (25
mg/kg) for 40 days (AN15368: n=4; untreated: n=9 mice; naïve n=3). Parasite load in
skeletal muscle after immunosuppression with cyclophosphamide was assessed by qPCR.
Untreated animals in **(e)** were terminated and tissue samples collected before the end of
the experiment due to the high susceptibility to *T. cruzi* infection and the lethal pathology

developed by IFN-gamma deficient animals. Red filled symbols denote animals in which parasites were detected by fresh blood smears or hemoculture. Data are presented as mean values +/- SEM. Dotted lines represent the cutoff in the qPCR assays. For **b,d** and **e**: Mann Whitney test, two tailed; for **c**: Unpaired t test, 95% confidence interval. *=p 0.05; **=p 0.01; ***=p 0.001.

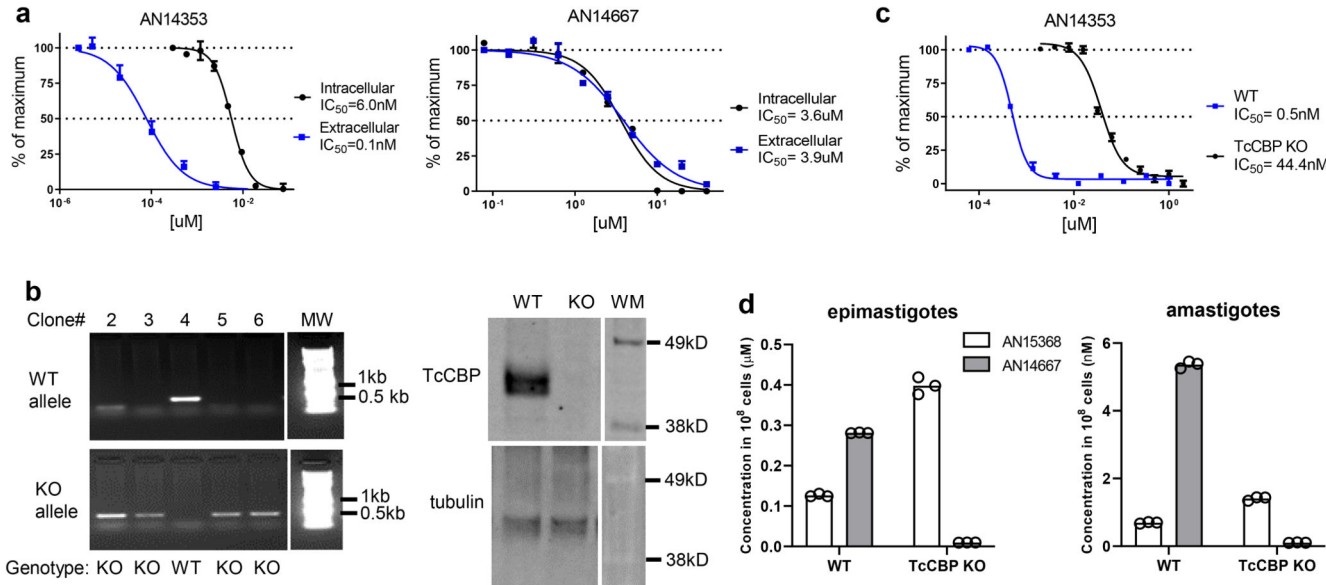

**Fig. 2. *T. cruzi* serine carboxypeptidases activate lead benzoxaborole compounds.**
**(a)** IC50 of parent compound AN14353 and predicted peptidase cleavage product AN14667 on intracellular and extracellular amastigotes. Three replicates were performed at each concentration: assays were repeated > 3 times. **(b)** Generation of serine carboxypeptidase (CBP) KO lines in epimastigotes. WT and KO alleles was amplified using gene-specific primers respectively (left panel; see Methods). Clones with only KO but not WT allele are considered to be KO lines. Western blot validated the absence of TcCBP protein in the KO line using a TcCBP-specific antibody and tubulin detection as a control (right panel; TcCBP protein and tubulin were detected in different gels). **(c)** Impact of CBP disruption on activity of AN14353 on extracellular amastigotes **(d)** Uptake and conversion of AN15368 to CBP cleaved product AN14667 in WT and CBP-disrupted epimastigotes (top; treated with 10 uM AN15368) and amastigotes (bottom; treated with 50 nM AN15368) of *T. cruzi*. Data are presented as mean values +/- SEM; **b** and **c**: n=3 biological replicates; **d**: data are presented as individual values of 3 technical replicates.

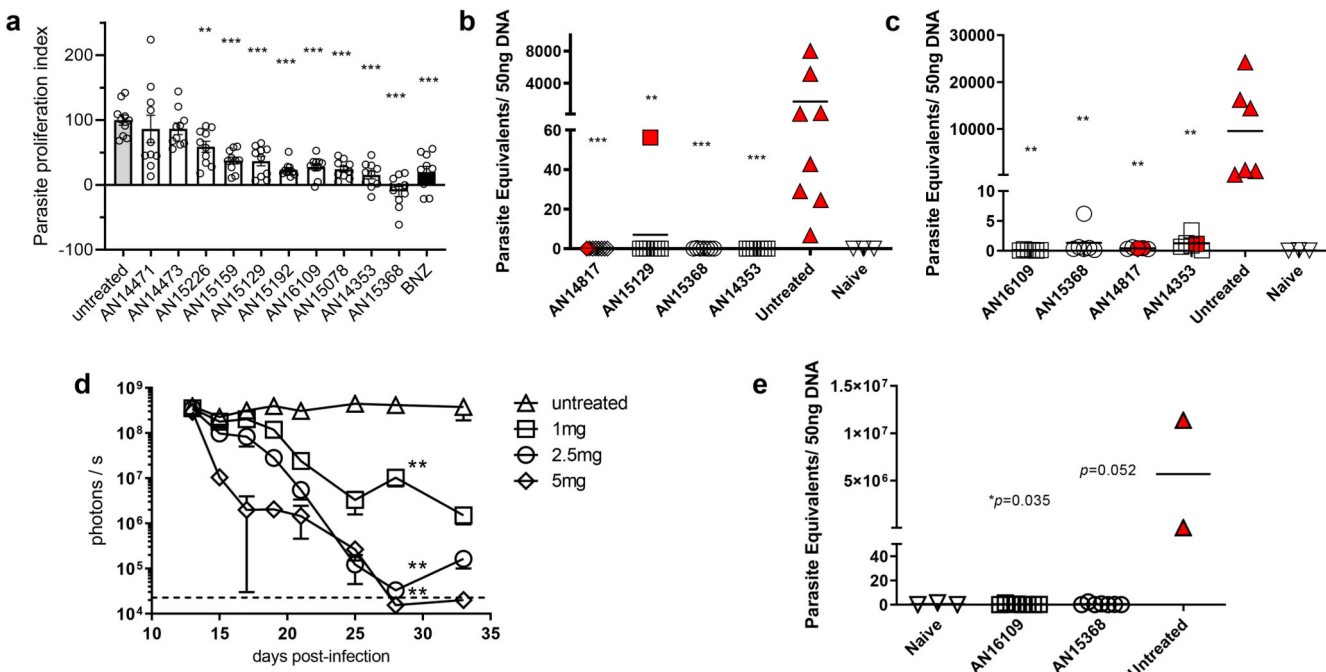

**Fig. 3. AN15368 cures *T. cruzi* infection in mice.**

**(a)** Fluorescence intensity of C57BL/6 mice infected in the footpads with $2.5 \times 10^5$ tdTomato-expressing *T. cruzi* was measured by in vivo imaging directly before (2 dpi) and 2 days after (4 dpi) a single oral dose of the compounds (50 mg/kg). The parasite proliferation index after treatment was calculated as described in the Materials & Methods section. (n=10 individual feet). **(b)** A daily oral treatment of the indicated compounds (10mg/kg) was administered for 40 days to C57BL/6 mice acutely infected with *T. cruzi* beginning at 17 dpi. Mice were immunosuppressed at the end of the treatment and samples were collected for parasitemia, hemoculture and qPCR of skeletal muscle. Dotted line represents the background level from naïve mice (untreated, AN14817 and AN15129: n=8; AN15368 and AN14353: n=7; naive: n=3 mice). **(c)** Acutely infected mice (20 dpi) were treated orally with 10 mg/kg of compounds for 20 days. Parasite load in skeletal muscle was quantified by qPCR following immunosuppression at the end of the treatment. Dotted line represents the cutoff for the qPCR assay (untreated, AN16109, AN15368 and AN14353: n=6; AN14817: n=5; naive: n=3 mice). **(d)** On days 13-33 post infection, oral doses of AN15368 were administered at the indicated concentrations to hairless mice infected i.p. with $5 \times 10^4$ Luciferase-expressing *T. cruzi*. Bioluminiscence signals from whole mouse ventral images were acquired and quantified throughout the treatment by in vivo imaging (untreated =7 mice; treated = 5 mice). **(e)** C57BL/6 mice infected with $10^4$ *T. cruzi* i.p. were daily treated for 40 days, starting at 15 dpi with AN16109 or AN15368 (both at 2.5 mg/kg). Parasite load was determined by qPCR of skeletal muscle samples after immunosuppression and blood samples were analyzed by light microscopy and hemoculture (AN16109: n=8; AN15368: n=7; naive: n=3; untreated: n=2 mice). Red symbols denote animals in which parasites were detected by microscopy and/or hemoculture. Data are presented as mean values +/- SEM. Asterisks represent the statistical significance of the difference between

each treated group and the untreated control. For **a**, **b** and **c:** Mann Whitney test, two tailed; for **e**: Unpaired t test, 95% confidence interval. *=p 0.05; **=p 0.01; ***=p 0.001.

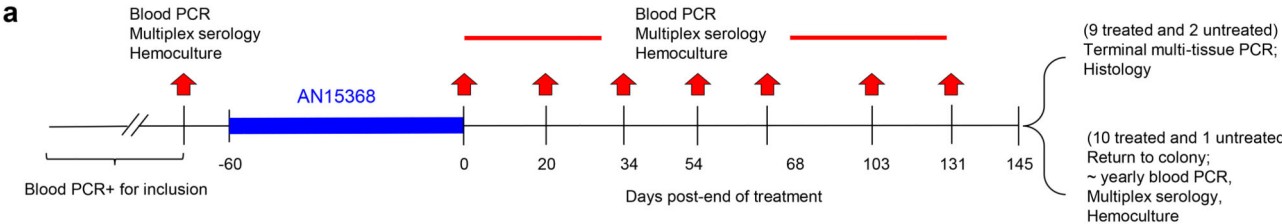

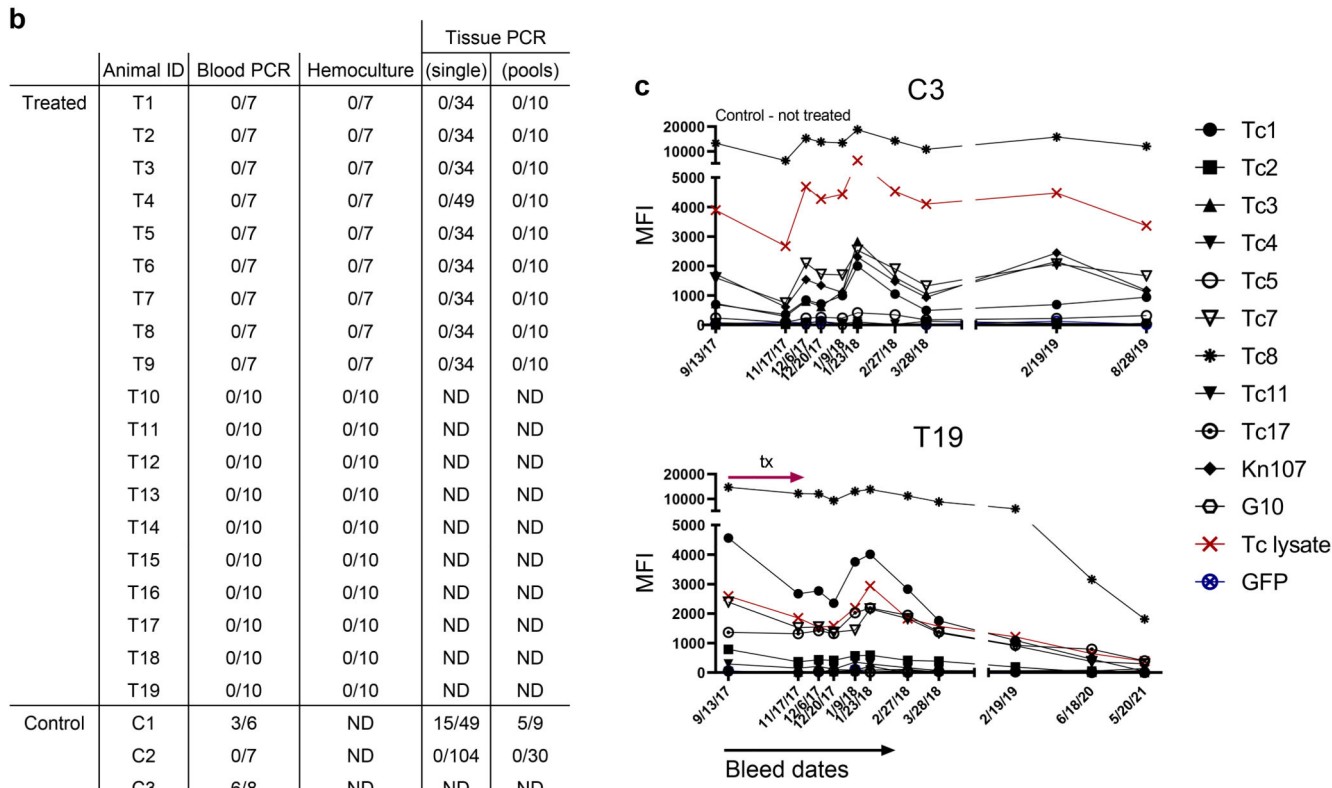

**Fig. 4. AN15368 uniformly cures chronically infected rhesus macaques.**
**(a)** Structure of drug trial with indicated treatment period and sampling dates. **(b)** Summary of post-treatment detection of infection using PCR of parasite DNA in blood or tissue (either from individual tissue samples or pools of 5 individual samples/determination – see M&M and Table S8-S10 for details) and hemoculture in **drug-treated** (T1-T19) and untreated **controls** (C1-3) macaques. Samples were taken at the 7 time points indicated in (a) and yearly thereafter for animals T1-T19, that were not euthanized and thus not tissue sampled for PCR. Fractions in table (e.g. "0/7") indicate number of positive results per total number of samples analyzed. **(c)** Representative pre- and -post treatment IgG responses to recombinant *T. cruzi* proteins in control animal C3 and treated animal T19 (see Methods for identification and Extended data 3 for remaining animals). ND = not done.

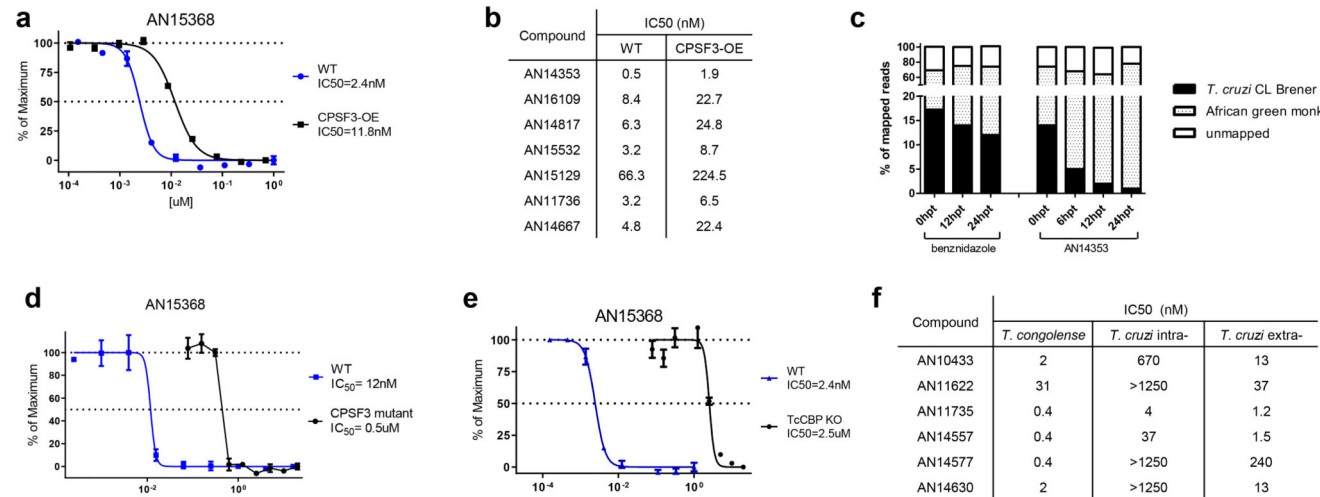

**Fig. 5. AN15368 is activated by a *T. cruzi* serine carboxypeptidase and targets CPSF3.**
**(a)** Overexpression of CPSF3 increases resistance to AN15368(CPSF3-OE: n=3 biological replicates; WT: n=2 biological replicates), **(b)** as well as to related benzoxaboroles. All assays were carried out in intracellular amastigotes. **(c)** Suppression of parasite mRNA relative to host cell (African Green Monkey (Vero)) mRNA by exposure to AN14353 as indicated by the decline in the fraction of sequence reads mappable to parasite genome relative to host cell genome. **(d)** Engineered Asn to His mutation at amino acid position 232 of CPSF3 conveys resistance to AN15368 by intracellular amastigotes (n=3 biological replicates) **(e)** Disruption of CBPs leads to a 1000X increased resistance to AN15368 in intracellular amastigotes(TcCBP KO: n=3 biological replicates; WT: n=2 biological replicates). **(f)** Some AN15368 analogues lacking activity to intracellular *T. cruzi* (intra-) are effective killers of the *T. congolense* (extracellular) and of *T. cruzi* extracellular amastigotes (extra-). Data are presented as mean values +/- SEM.

*Nat Microbiol.* Author manuscript; available in PMC 2022 October 03.