## [Peer Review File · Nature microbiology]

Peer Review Information

Journal: Nature Microbiology

Manuscript Title: Discovery of AN15368, An Orally Active Benzoxaborole as a Treatment for Chagas Disease

Corresponding author name(s): Dr Rick Tarleton

Reviewer Comments & Decisions:

Decision Letter, initial version:
--

2nd March 2022

Dear Dr Tarleton,

Thank you for your patience while your manuscript "Discovery of AN15368, An Orally Active Benzoxaborole as a Potential Treatment for Chagas Disease" was under peer-review at Nature Microbiology. It has now been seen by 4 referees, whose expertise and comments you will find at the of this email. You will see from their comments below that while they find your work of interest, some important points are raised. We are very interested in the possibility of publishing your study in Nature Microbiology, but would like to consider your response to these concerns in the form of a revised manuscript before we make a final decision on publication.

In particular, you will see that the referees feel the findings need be presented more clearly and explained better. They also have some technical questions regarding the NHP model, and feel the novelty of this model for Chagas Disease should be better highlighted. Specifically, referee #1 feels that the paper needs more discussion or explanation, and is currently hard read in place. This referee also has some questions regarding data presentation in the different figures and tables. Referee #2 has a number of concerns with the presentation that must be addressed before publication. This referee feels the figure legends are too brief, there is no description of experimental reproducibility for any of the figures showing in vitro data, and that Figure 4 is incomprehensible and needs to be edited. Referee #2 also states that the discussion is very narrow and the impact of the findings should be highlighted better. Referee #3 feels some aspects of the work need clarification, specifically regarding the mouse experiments, dormant cells and resistance. This referee also has some questions regarding the NHP model, and asks "Is there a statistical test that can give a level of confidence in the incidence of real cure over possible spontaneous cure in the treated group?". Referee #3 also feels that some more statistical analysis would be helpful on the cardiac issues identified. Referee #4 feels that the lack of positive controls, such as the standard of care (e.g. SOC, Benznidazole or Nifurtimox) makes it difficult for the reader to appreciate the significance of the results achieved in some of those experiments (e.g. Fig 1B-D and Fig 3D), says the preclinical safety assessment is perfunctory and the in vitro safety data should be provided as supplemental information. Referee #4 also states that there are no safety margins estimate to help the reader appreciate the relative safety of this novel chemical entity. The referee also has questions regarding the duration of treatment of NHP and the dose used. The referee also feel that some insights should be provided into the efficacy of the standard of care Benznidazole in this NHP model. Editorially, we would strongly encourage you to add such data if available. Referee #4 also feels that small number of control animals do not allow a robust analysis of the interesting heart histopathology findings. The referee furthermore says that more insights into potential drug resistance liability through drug pressure selection could reveal clinically relevant drug resistance mutations in TcCPSF3 and/or TcCBP. Editorially, we would again encourage you to add such

data to the manuscript if available. The rest referees' reports are clear and the remaining issues should be straightforward to address.

As the study uses NHPs, please ensure that you mention ethics oversight and have followed the ARRIVE guidelines (<https://arriveguidelines.org/>).

If you have not done so already please begin to revise your manuscript so that it conforms to our Article format instructions at <http://www.nature.com/nmicrobiol/info/final-submission/>

The usual length limit for a Nature Microbiology Article is six display items (figures or tables) and 3,000 words. We have some flexibility, and can allow a revised manuscript at 3,500 words, but please consider this a firm upper limit. There is a trade-off of ~250 words per display item, so if you need more space, you could move a Figure or Table to Supplementary Information.

Some reduction could be achieved by focusing any introductory material and moving it to the start of your opening 'bold' paragraph, whose function is to outline the background to your work, describe in a sentence your new observations, and explain your main conclusions. The discussion should also be limited. Methods should be described in a separate section following the discussion, we do not place a word limit on Methods.

Nature Microbiology titles should give a sense of the main new findings of a manuscript, and should not contain punctuation. Please keep in mind that we strongly discourage active verbs in titles, and that they should ideally fit within 90 characters each (including spaces).

Please include a data availability statement as a separate section after Methods but before references, under the heading "Data Availability". This section should inform readers about the availability of the data used to support the conclusions of your study. This information includes accession codes to public repositories (data banks for protein, DNA or RNA sequences, microarray, proteomics data etc...), references to source data published alongside the paper, unique identifiers such as URLs to data repository entries, or data set DOIs, and any other statement about data availability. At a minimum, you should include the following statement: "The data that support the findings of this study are available from the corresponding author upon request", mentioning any restrictions on availability. If DOIs are provided, we also strongly encourage including these in the Reference list (authors, title, publisher (repository name), identifier, year). For more guidance on how to write this section please see:

<http://www.nature.com/authors/policies/data/data-availability-statements-data-citations.pdf>

To improve the accessibility of your paper to readers from other research areas, please pay particular attention to the wording of the paper's opening bold paragraph, which serves both as an introduction and as a brief, non-technical summary in about 150 words. If, however, you require one or two extra sentences to explain your work clearly, please include them even if the paragraph is over-length as a result. The opening paragraph should not contain references. Because scientists from other sub-disciplines will be interested in your results and their implications, it is important to explain essential but specialised terms concisely. We suggest you show your summary paragraph to colleagues in other fields to uncover any problematic concepts.

If your paper is accepted for publication, we will edit your display items electronically so they conform to our house style and will reproduce clearly in print. If necessary, we will re-size figures to fit single or double column width. If your figures contain several parts, the parts should form a neat rectangle when assembled. Choosing the right electronic format at this stage will speed up the processing of your paper and give the best possible results in print. We would like the figures to be supplied as vector files - EPS, PDF, AI or postscript (PS) file formats (not raster or bitmap files), preferably generated with vector-graphics software (Adobe Illustrator for example). Please try to ensure that all figures are non-flattened and fully editable. All images should be at least 300 dpi resolution (when figures are scaled to approximately the size that they are to be printed at) and in RGB colour format. Please do not submit Jpeg or flattened TIFF files. Please see also 'Guidelines for Electronic Submission of Figures' at the end of this letter for further detail.

Figure legends must provide a brief description of the figure and the symbols used, within 350 words, including definitions of any error bars employed in the figures.

When submitting the revised version of your manuscript, please pay close attention to our [href="https://www.nature.com/nature-research/editorial-policies/image-integrity">Digital Image Integrity Guidelines.](https://www.nature.com/nature-research/editorial-policies/image-integrity) and to the following points below:

Please include a statement before the acknowledgements naming the author to whom correspondence and requests for materials should be addressed.

Finally, we require authors to include a statement of their individual contributions to the paper -- such as experimental work, project planning, data analysis, etc. -- immediately after the acknowledgements. The statement should be short, and refer to authors by their initials. For details please see the Authorship section of our joint Editorial policies at

http://www.nature.com/authors/editorial_policies/authorship.html

* include a point-by-point response to any editorial suggestions and to our referees. Please include your response to the editorial suggestions in your cover letter, and please upload your response to the referees as a separate document.

* ensure it complies with our format requirements for Letters as set out in our guide to authors at www.nature.com/nmicrobiol/info/gta/

* state in a cover note the length of the text, methods and legends; the number of references; number and estimated final size of figures and tables

* resubmit electronically if possible using the link below to access your home page:

[Redacted]

*This url links to your confidential homepage and associated information about manuscripts you may have submitted or be reviewing for us. If you wish to forward this e-mail to co-authors, please delete this link to your homepage first.

Please ensure that all correspondence is marked with your Nature Microbiology reference number in the subject line.

Nature Microbiology is committed to improving transparency in authorship. As part of our efforts in this direction, we are now requesting that all authors identified as 'corresponding author' on published papers create and link their Open Researcher and Contributor Identifier (ORCID) with their account on the Manuscript Tracking System (MTS), prior to acceptance. This applies to primary research papers only. ORCID helps the scientific community achieve unambiguous attribution of all scholarly contributions. You can create and link your ORCID from the home page of the MTS by clicking on 'Modify my Springer Nature account'. For more information please visit www.springernature.com/orcid.

We hope to receive your revised paper within four weeks. If you cannot send it within this time, please let us know.

Yours sincerely,
[Redacted]

Reviewer Expertise:

Referee #1: Trypanosome cellular and molecular biology
Referee #2: Anti-parasitic drug discovery and development

Referee #3: Antiparasitic drugs

Referee #4: Anti-parasitic drug discovery and development

Reviewers Comments:

Reviewer #1 (Remarks to the Author):

Lack of much in the way of discussion or explanation. Makes less accessible to many readers. Appreciate the need for brevity, but this is a hard read in places

Why color in F3 when not in F1 with similar data type.

Aware immunosuppression tips in favour of parasite, but what is impact in immunocompetent mice?

F4 - why include T10 onwards in panel B? When were these analyses done? What are pools?
I do not understand the reason for the fusion of the antigens used in panel C, and moreover we have T10 etcetera referring to an animal, and Tc1 etcetera referring to an antigen. I'm a bit slow but very confusing, both to interpret as well as to understand the rationale in places.

I am not sure that loss of antigen reactivity is really a good standard, with our new understanding of latency. While may have been in the past, perhaps needs to be more circumspect.

The potential for resistance emerging, due to the long dosing regime should at least be mentioned.

Reviewer #2 (Remarks to the Author):

The manuscript by Padilla et al reports on the very exciting discovery of an orally active benzoxaborole that is curative for *T. cruzi* infection in both mice and non-human primates, and appears to have excellent potential to progress to preclinical development for the treatment of Chagas disease. This is a highly significant study because it has been very difficult to identify new drugs for the treatment of Chagas. The current drugs benznidazole and nifurtimox are so toxic that many adults can't tolerate a full course of treatment. Yet drug discovery has proven difficult because the parasite invades many types of tissues, and also has a dormant form. It has been difficult to translate good in vitro results to in vivo efficacy at the level of curative. The authors of this current study circumvent that by performing in vivo mouse studies on a larger number of their potential compounds during lead optimization. But while many initially promising series that have been discovered by other groups over the years eventually have failed to match the efficacy of benznidazole in providing cures in mice (e.g. some parasite suppression is observed but not cures), the benzoxaborole series described herein appears to be the first to achieve this formidable goal. The package of studies assessing the compounds described in the manuscript is strong and includes in vitro analysis of activity on parasites, extensive in vivo analysis in mice and non-human primates, and appropriate ADME data including pharmacokinetics. The non-human primate studies were done in a colony of rhesus macaques that had acquired natural infections in their habitat in the US, and their infection mimics the diversity seen in human infections. So the success of the lead compound in this model really strengthens the conclusions that the compounds described can be curative. The manuscript finally also describes MOA studies showing that at least in part these compounds target CPSF3, which is similar to what was

published for similar compounds in *T. brucei*. Overall this is a very comprehensive manuscript and its findings are of the highest impact.

However I have a number of concerns with the presentation that must be addressed before publication. I understand they were trying to write a very large story in a compact space, but this has been attempted to the detriment of clarity.

1) Figure legends are so brief that one can't understand the studies. All figure legends need to be reconsidered with this point in mind. But to give some examples. Fig. 2B. It's not enough to say this gel shows a KO. This figure has to be explained. What are we looking at? Fig 2C. According to the methods the genetic studies were done in epimastigotes. The figure legend needs to specify this detail. Fig 2D, y-axis concentration in what? uM based on some intracellular conversion, or concentration in some extract? This needs to be explained. And why show epimastigotes when all other studies are in amastigotes. Fig. 5, ditto above comments, we need more explanation in the legends.

2) No description of experimental reproducibility is provided for any of the figures showing in vitro data. The meaning of error bars is not provided in figure legends. Many tables contain single EC50 values with no indication that the data were repeated. I wouldn't expect every SAR point for every compound to be repeated, but I would expect that EC50 data for key compounds had been determined in multiple independent experiments. In contrast error analysis is well done and appropriate for in vivo mouse studies.

3) Figure 4. This figure is incomprehensible. The table in part B doesn't explain what the numbers mean. For example 0/7, what does the denominator represent. Figure 4C is so small that I can't read it even zoomed out to the largest setting on my computer monitor. Thus I can't evaluate these data. This figure need to be entirely redone so that the message gets across to the reader, and so that the data can be evaluated. Frankly I think Figure 4C could be sent to the supplement, and maybe just a representative curve for one animal shown in the main manuscript.

4) The discussion is disappointingly narrow in its focus. The manuscript describes a really exciting discovery. The discussion could do a lot better job of bringing out the impact of their findings.

More minor points.

1) Page 4 line 2, S9 microsomal assays are mentioned but no reference to the relevant supplemental table is provided.

2) Significant figures wander all over the place from 2-4 in the same table. Please be consistent.

3) if the authors have any comparative data to benznidazole run in the same in vivo study, it would be nice to incorporate that data.

Reviewer #3 (Remarks to the Author):

This manuscript reports a pre-clinical package of studies that bring the benzoxaborole AN15368 to the position of leading candidate to pursue against Chagas disease, a neglected tropical disease for which the current therapeutic options are unsatisfactory and against which finding new medications has been difficult. The benzoxaborole class has proven itself to be of remarkable utility against a range of protozoan pathogens and one molecule of the class is very close to registration for use against human African trypanosomiasis. A second is well advanced against Animal African trypanosomiasis and

candidates are being investigated against malaria, toxoplasmosis, leishmaniasis, cryptosporidiosis and other protozoal infections.

The overall package of work is comprehensive. In addition to demonstrating efficacy against *T. cruzi* parasites in various in vitro and in vivo studies, including non-human primates, a mode of action has been studied and proposed and resistance mechanisms investigated with a clear pathway of a mode of action involving uptake into host cells, then parasites, followed by processing of a prodrug precursor to accumulated active product that inhibits RNA processing through CPSF3. Pharmacokinetic and toxicity testing in preclinical models has been advanced.

There are some aspects of the work I feel need clarification.

1. Please state which is the other heart disease causing microbe. Is it streptococcus? There are others too though, e.g. African trypanosomes so clarity is needed (or the phrase needs amending)
2. With regard to dosing. I found the mouse experiments somewhat difficult to follow. It is desirable to find an agent that can be given for reduced time compared with current therapies. A 40 day treatment protocol was used for WT mice, which is standard. However, a 20 day regimen was tested too to help discriminate better and worse compounds. Were even shorter regimens considered for the best compounds? It would be useful in the text to offer direct comparisons to what happens with benznidazole treatment in the same models. If shorter treatments did not cure, is this likely to be due to presence of low numbers of dormant parasites? If these relapse later than with say benznidazole treated mouse infections is there anything in the comparative PK that can cause that?
3. The cure of mice with the 40 day dosing leads the authors to conclude that dormant cells appear not to be an issue (over that time frame). They also make a clear case that the multiple and protracted dosing of these compounds compared with single dose treatment to cure African trypanosomes (the compound in development for animal African trypanosomes is of the same cleavable valine ester class) is likely due to the intracellular, multi-organ localisation of *T. cruzi* vs African trypanosomes which are extracellular. However, as the authors do have good assay systems that have shown how dormant cells can be refractory to benznidazole have those same assays been applied to AN15368? These data would be very helpful in helping frame the discussion around dormant cells
4. With respect to the variable copy number of the TcCLB array across strains. Does this have implications for either variability in parasite strain specific responses, or in the possible relative risk of resistance emerging?
5. The NHP model takes advantage of a colony of naturally infected macaques, rather than controlled infection conditions. For ethical reasons there are grounds to have this relatively uncontrolled model. However, there were a number of issues that do leave a degree of uncertainty in these experiments. Firstly, the issue of apparent spontaneous cure in one untreated monkey leads to some questions about the model, although the authors have addressed this in the Discussion. Is there a statistical test that can give a level of confidence in the incidence of real cure over possible spontaneous cure in the treated group?
7. Was the fact that animals returning to the breeding colony remained Chagas disease free a surprise given that the parasites seem to transmit freely there?

8. On page 8 line 20, giving the animals food treats is mentioned. How often and what were those food treats? If ad hoc rather than scheduled this should be discussed either here or in methods. In methods, some of the primate PK work used pumpkin slurry. IS this a standard procedure known not to influence PK?
9. Some more stats would be helpful on the cardiac issues identified since 2 treated (out of 19) versus one untreated (out of 3) would indicate that the incidence in treated is either less or certainly of no significant difference than in untreated, but it would be good to show the workings.
10. The Discussion, page 9 line 32 I would change "the" apparent target to "an" apparent target as polypharmacology of the compound can't be ruled out at this point. Same on line 5 of page 10.

Reviewer #4 (Remarks to the Author):

The manuscript entitled "Discovery of AN15368, An Orally Active Benzoxaborole as a Potential Treatment for Chagas Disease" by Padilla et al. describes the discovery and preclinical characterization of the drug candidate AN15368 belonging to the class of benzoxaborole for the treatment of Chagas disease. Chagas disease remains a significant Global Health R&D priority, as the available therapies, to manage and prevent the disease, are broadly viewed as inadequate. Novel and better therapeutics to treat Chagas are urgently needed. Given the scarcity of Chagas drug discovery efforts, the results reported here are thus significant advances in the field, in particular as it relates to the use of a Nonhuman Primate model of Chagas disease which has not been previously used to support drug discovery.

The manuscript provides a detailed description of the medicinal chemistry optimization from the previously reported lead AN4169 (reference 4) to the ultimate candidate AN15368. The authors provide a compelling dataset supporting the successful multi-parameters optimization of this chemical series, notwithstanding a more in depth description of this work would be valuable, although more appropriate for a more specialized medicinal chemistry journal.

Following on the previously published finding that *T. cruzi* carboxypeptidase can cleave and activate ester containing pro-drug (reference 21), the authors provide strong genetic evidence that the serine carboxypeptidase TcCBP is required for the conversion of AN15368 to its active carboxylic acid metabolite AN14667.

The authors provide evidence that AN15368 is potently active against *T. cruzi* in a variety of in vitro and in vivo assays. Although, the lack of positive controls, such as the standard of care (e.g. SOC, Benznidazole or Nifurtimox) makes it difficult for the reader to appreciate the significance of the results achieved in some of those experiments (e.g. Fig 1B-D and Fig 3D). The preclinical safety assessment is perfunctory and the in vitro safety data should be provided as supplemental information. The safety study in rodent is only for seven days and the authors do not establish the potential target organs for this drug class as no histopathology is reported for any of the doses tested. The reader cannot relate the provided exposure data at the NOAEL (AUC =30,000 ng*hr/mL reported page 6 line 36, is this total of free AUC?) to the potential exposure anticipated to be required for efficacy in humans. Finally, there are no safety margins estimate to help the reader appreciate the relative safety of this novel chemical entity. Note that based on monkey exposure data provided,

these margins appears to be potentially small (e.g. < 3).

The most interesting and novel part of this manuscript reports the efficacy of AN15368 in a Non-Human Primates (NHP) Chagas disease model. Because these animals are naturally and chronically infected with a broad range of *T. cruzi* genetic types (or DTU) and display some of the pathology associated with chronic Chagas disease, this model could potentially be more accurately reflecting the clinical situation of chronic indeterminate Chagas patients. The authors decided on a 60-day treatment regimen without a strong rationale other than to say that this is the standard treatment regimen for patients treated with benznidazole. The dose of 30 mg/kg used is also poorly justified. Why did the authors choose the target of > 1 ng/ml for 20-24 hrs? What is the PK/PD model/hypothesis that they used to make these decisions? Why did they believe that achieving this concentration, regardless of whether this is protein-bound of free plasma concentration, would be efficacious? Notwithstanding, the authors provide compelling evidence that AN15368 treatment for 60 days at 30 mg/kg is well tolerated and the compound is able to clear the infection in almost all animals in both blood and tissues. Moreover, monitoring of antibodies suggest that all animals seroconvert after treatment, as anti-*T. cruzi* antibodies decline over a 42 months period after the end of treatment. This is very important because seroconversion is a critical endpoint for clinical trials in Chagas patients. Unfortunately, this study has a number of shortcomings that do not allow the reader to draw any conclusions with respect to the potential advantages of AN15368 to treat the key target population of chronic indeterminate Chagas patients. First, the authors should provide insights into the efficacy of the standard of care Benznidazole in this NHP model. Second, a more thorough assessment of the histopathology in chronically *T. cruzi* infected NHP would help assess whether one could expect a reversal of some of the pathologies associated with chronic infection in Chagas patients. Only three untreated animals were analyzed, and one animal cured spontaneously without treatment, this small number of control animals do not allow a robust analysis of the interesting heart histopathology findings reported in data S1 table. Do the animals treated with AN15368 resolve most of these symptoms upon parasite clearance and seroconversion? The authors with this current study do not address this important question, and it would be a very significance advance to the field if a larger negative control cohort had been used.

In the final section of this manuscript, the authors show that, like other previously described oxaboroles developed for the treatment of Sleeping Sickness (refs 7, 19 and 20), AN15368 likely inhibits the *T. cruzi* Cleavage and Polyadenylation Specificity Factor (CPSF3) enzyme involved in the regulation of mRNA processing and maturation. The authors provide genetic evidence that overexpression of CPSF3 results in significant but moderate level of drug resistance to AN15368. In support of this hypothesis, the authors provide data showing a specific decrease in *T. cruzi* mRNA expression upon AN15368 treatment. Collectively these data are consistent with previous reports but do not provide direct and more definitive evidence that CSPF3 is the actual direct target of AN15368. It is notably regrettable that the authors do not provide more insights into potential drug resistance liability through drug pressure selection that could reveal clinically relevant drug resistance mutations in TcCPSF3 and/or TcCBP.

Overall, this manuscript provides a good overview of the discovery of a novel agent for the treatment of Chagas disease and provide the first description of a NHP model for Chagas disease. This is an important report and addressing several of the issues highlighted above would make this manuscript more attractive to the general readership of Nature Microbiology.

Minor comments:

1. The section page 2 lines 21-25 is a very long sentence and could be rewritten.
2. Adding short headings for sections would help the reader following the paper.
3. Page 3. Lines 1-2 suggest that metabolic stability tracked with clogD but most compounds reported in Table S2 have low clearance regardless of clogD values.
4. Page 4 line 2 : Change "Several" to 4 (as only 4 of the 10cpds have 10 ug*hr/ml AUC)
5. Page 4 lines 6-12: hasn't this data been published before?
6. Page 5 line 26 – "greatest" does not seem necessary here.
7. Fig.3D. it would be great to see the images as supplemental data
8. Page 6: line 35-36: Rat PK: FigS4. i.v PK is missing, also provide the Cmax, AUC, F in tabular form. Provide therapeutic window with respect to doses required for showing efficacy in mouse model.
9. Page 6: line 43-44: Monkey PK: Fig. S5. i.v PK missing, also provide the Cmax, AUC, F in tabular form
10. Page 9 lines 16-23: it is intriguing to see some of the analogs active against extracellular but not intracellular amastigotes. Could this be due to permeability issues? Can the authors provide measurement of permeability of these compounds? Alternatively, is it possible that AN15368 is not cleaved efficiently by human host (but not in other non-human species) and that could be the reason for good activity against Tc and no other trypanosoma species (Page 10 lines 8-15)
11. Supplementary info: Most references appear to be cited incorrectly or perhaps the references for the supplemental information are missing.
12. In vitro assays section; please check references (6 describes Sleeping sickness; 24 describes Crypto)
13. In vivo assays section: Cure assay: check the references (ref 25 is for apicomplexan parasites and ref 23 describes toxoplasma). More details on the in vivo experimental protocols would be helpful. (e.g. beginning of the treatment, number of days treatment was given, immunosuppression regimen, imaging schedule).
14. Page 3: CPSF 3 cloning: Check references: ref 9 describes AAT; RNA seq check references. 28, 29 20 31– ref 28 describes spontaneous Chagas;
15. Page 4: 32 ref is missing in references list.
16. Page 5, 3rd para – PK reference is missing.
17. Protocol on DTU typing and growth inhibition testing are not provided – kindly include in supplemental information.
18. Monkey PK: Fig. S5. i.v PK is missing and please provide the Cmax, AUC, F in tabular form.

Author Rebuttal to Initial comments

Point-by-point response to Reviewers Comments (*author's responses in italics*):

Reviewer #1:

Why color in F3 when not in F1 with similar data type.

Color added to F1 to match F3

Aware immunosuppression tips in favour of parasite, but what is impact in immunocompetent mice?

We have added a line to the M&M to better indicate the purpose of the immunosuppression.

F4 - why include T10 onwards in panel B? When were these analyses done? What are pools? I do not understand the reason for the fusion of the antigens used in panel C, and moreover we have T10 etcetera referring to an animal, and Tc1 etcetera referring to an antigen. I'm a bit slow but very confusing, both to interpret as well as to understand the rational in places.

Fig 4B includes data on post-treatment PCR and hemoculture for all animals, including animals T10-19 (hence the need for inclusion of all animals in the Table. The figure legend has been modified to better describe animal and antigen numbering as well as the sampling protocol. Materials and Methods have been modified to include justification for using fusion antigens in antibody assays.

I am not sure that loss of antigen reactivity is really a good standard, with our new understanding of latency. While may have been in the past, perhaps needs to be more circumspect.

*Our understanding of latency in *T. cruzi* is that individual parasites are temporarily dormant (latent) but that the infection overall is not latent (there is likely a (nearly) continuous presence of both actively dividing and dormant (latent) parasites in infected (including chronically infected) hosts. In keeping with that, it is the overwhelming finding (decades of work in multiple species) that chronically infected hosts remain antibody positive and that drug cure results in the decline or loss of these antibodies over time. We are not aware of any data on latency that changes this understanding or interpretation.*

The potential for resistance emerging, due to the long dosing regimen should at least be mentioned.

Comment added to discussion

Reviewer #2:

The manuscript by Padilla et al reports on the very exciting discovery of an orally active benzoxaborole that is curative for *T. cruzi* infection in both mice and non-human primates, and appears to have excellent potential to progress to preclinical development for the treatment of Chagas disease. This is a highly significant study because it has been very difficult to identify new drugs for the treatment of Chagas. The current drugs benznidazole and nifurtimox are so toxic that many adults can't tolerate a full course of treatment. Yet drug discovery has proven difficult because the parasite invades many types of tissues, and also has a dormant form. It has been difficult to translate good in vitro results to in vivo efficacy at the level of curative. The authors of this current study circumvent that by performing in vivo mouse studies on a larger number of their potential compounds during lead optimization. But while many initially promising series that have been discovered by other groups over the years eventually have failed to match the efficacy of benznidazole in providing cures in mice (e.g. some parasite suppression is observed but not cures), the benzoxaborole series described herein appears to be the first to achieve this formidable goal. The package of studies assessing the compounds described in the manuscript is strong and includes in vitro analysis of activity on parasites, extensive in vivo analysis in mice and non-human primates, and appropriate ADME data including pharmacokinetics. The non-human primate studies were done in a colony of rhesus macaques that had acquired natural infections in their habitat in the US, and their infection mimics the diversity seen in human infections. So the success of the lead compound in this model really strengthens the conclusions that the compounds described can be curative. The manuscript finally also describes MOA studies showing that at least in part these compounds target CPSF3, which is similar to what was published for similar compounds in *T. brucei*. Overall this is a very comprehensive manuscript and its findings are of the highest impact.

However I have a number of concerns with the presentation that must be addressed before publication. I understand they were trying to write a very large story in a compact space, but this has been attempted to the detriment of clarity.

1) Figure legends are so brief that one can't understand the studies. All figure legends need to be reconsidered with this point in mind. But to give some examples. Fig. 2B. It's not enough to say this gel shows a KO. This figure has to be explained. What are we looking at?

Fig 2C. According to the methods the genetic studies were done in epimastigotes. The figure legend needs to specify this detail. Fig 2D, y-axis concentration in what? uM based on some intracellular conversion, or concentration in some extract? This needs to be explained. And why show epimastigotes when all other studies are in amastigotes. Fig. 5, ditto above comments, we need more explanation in the legends.

Figure legends have been modified to incorporate these details; amastigote data added to Fig 2.

2) No description of experimental reproducibility is provided for any of the figures showing in vitro data. The meaning of error bars is not provided in figure legends. Many tables contain single EC50 values with no indication that the data were repeated. I wouldn't expect every SAR point for every compound to be repeated, but I would expect that EC50 data for key compounds had been determined in multiple independent experiments. In contrast error analysis is well done and appropriate for in vivo mouse studies.

This information has been added.

3) Figure 4. This figure is incomprehensible. The table in part B doesn't explain what the numbers mean. For example 0/7, what does the denominator represent. Figure 4C is so small that I can't read it even zoomed out to the largest setting on my computer monitor. Thus I can't evaluate these data. This figure need to be entirely redone so that the message gets across to the reader, and so that the data can be evaluated. Frankly I think Figure 4C could be sent to the supplement, and maybe just a representative curve for one animal shown in the main manuscript.

The figure legend has been modified to indicate the meaning in 4B and we have accepted the Reviewer's suggestion that results of a control and treated animal be include in 4C and the remainder moved to the supplemental data.

4) The discussion is disappointingly narrow in its focus. The manuscript describes a really exciting discovery. The discussion could do a lot better job of bringing out the impact of their findings.

The editors have graciously provided us additional space to expand the Discussion to emphasize the importance of the work, the attributes of the NHP model and to put in better perspective some of the findings in the Results section.

More minor points.

1) Page 4 line 2, S9 microsome assays are mentioned but no reference to the relevant supplemental table is provided.

Fixed

2) Significant figures wonder all over the place from 2-4 in the same table. Please be consistent.

Fixed

3) if the authors have any comparative data to benznidazole run in the same in vivo study, it would be nice to incorporate that data.

references added to published comparative data

Reviewer #3:

This manuscript reports a pre-clinical package of studies that bring the benzoxaborole AN15368 to the position of leading candidate to pursue against Chagas disease, a neglected tropical disease for which the current therapeutic options are unsatisfactory and against which finding new medications has been difficult. The benzoxaborole class has proven itself to be of remarkable utility against a range of protozoan pathogens and one molecule of the class is very close to registration for use against human African trypanosomiasis. A second is well advanced against Animal African trypanosomiasis and candidates are being investigated against malaria, toxoplasmosis, leishmaniasis, cryptosporidiosis and other protozoal infections.

The overall package of work is comprehensive. In addition to demonstrating efficacy against *T. cruzi* parasites in various in vitro and in vivo studies, including non-human primates, a mode of action has been studied and proposed and resistance mechanisms investigated with a clear pathway of a mode of action involving uptake into host cells, then parasites, followed by processing of a prodrug precursor to accumulated active product that inhibits RNA processing through CPSF3. Pharmacokinetic and toxicity testing in preclinical models has been advanced.

There are some aspects of the work I feel need clarification.

1. Please state which is the other heart disease causing microbe. Is it streptococcus? There are others too though, e.g. African trypanosomes so clarity is needed (or the phrase needs amending)

We assume this refers to the opening statement in the Introduction. There are many microbes (viral, bacterial, fungal, parasitic) that cause heart disease. We have narrowed the statement to "myocarditis" for clarity and provide a reference.

2. With regard to dosing. I found the mouse experiments somewhat difficult to follow. It is desirable to find an agent that can be given for reduced time compared with current therapies. A 40 day treatment protocol was used for WT mice, which is standard. However, a 20 day regimen was tested too to help discriminate better and worse compounds. Were even shorter regimens considered for the best compounds? It would be useful in the text to offer direct comparisons to what happens with benznidazole treatment in the same models. If shorter treatments did not cure, is this likely to be due to presence of low numbers of dormant parasites? If these relapse later than with say benznidazole treated mouse infections is there anything in the comparative PK that can cause that?

These are good questions. We have added reference to previous studies in which BZ was used in this reduced treatment period model, as requested. We are currently comparing in vitro and in vivo "relapse" times for BZ and AN15368. They do differ, but it is also clear that the relative resistance of dormant forms to both compounds prevent either from achieving cure in short-term treatment regimens (<20 days for AN15368 in this mouse model). In the case of both BZN and AN15368, all of the data suggest that it is the presence of dormant parasites that requires long-term treatment; each dose of (either) drug only kills actively dividing parasites but leaves dormant parasites largely untouched. The enhanced efficacy of AN15368 over BZN could be due to a number of factors, including the likely differential penetration of drug into all tissues where parasites may reside. We have also observed that BZN has both cytostatic and cytotoxic activity and we hypothesize that the cytostatic activity maintains parasites in a non-dividing state, making them "resistant" to the cytotoxic activities. AN15368 could be more effective than BZN because it lacks this property, although this hypothesis is still under investigation.

3. The cure of mice with the 40 day dosing leads the authors to conclude that dormant cells appear not to be an issue (over that time frame). They also make a clear case that the multiple and protracted dosing of these compounds compared with single dose treatment to cure African trypanosomes (the compound in development for animal African trypanosomes is of the same cleavable valine ester class) is likely due to the intracellular, multi-organ localisation of *T. cruzi* vs African trypanosomes which are extracellular. However, as the authors do have good assay systems that have shown how dormant cells

can be refractory to benznidazole have those same assays been applied to AN15368? These data would be very helpful in helping frame the discussion around dormant cells

See comment to 2. above – in short, and as supported by the minimum 20 day treatment period for efficient cures, AN15368 does not effectively kill dormant parasites. Comparisons to benznidazole using the assays we currently have at hand are difficult as benznidazole has cytostatic as well as cytotoxic activities and thus appears to hold parasites in a non-replicating (though not necessarily dormant) state while AN15368 is not cytostatic. But sorting this out – and determining the importance of these profiles on how these drugs are delivered, is complicated and we don't yet have the assays and data to tease it all apart.

4. With respect to the variable copy number of the TcCLB array across strains. Does this have implications for either variability in parasite strain specific responses, or in the possible relative risk of resistance emerging?

Great question. We observe no strain-specific variation in susceptibility to this class of benzoxaboroles (Table S4) and we as of yet have not been able to select for spontaneous TcCLB mutants, although this is still being investigated. It is also not clear that gene copy number correlates with peptidase activity, but this is also something we hope to determine in the future.

5. The NHP model takes advantage of a colony of naturally infected macaques, rather than controlled infection conditions. For ethical reasons there are grounds to have this relatively uncontrolled model. However, there were a number of issues that do leave a degree of uncertainty in these experiments. Firstly, the issue of apparent spontaneous cure in one untreated monkey leads to some questions about the model, although the authors have addressed this in the Discussion. Is there a statistical test that can give a level of confidence in the incidence of real cure over possible spontaneous cure in the treated group?

*We powered this test to detect cure in the drug-treated group, not for the control group. Spontaneous cure in any species is **extremely** rare (see references in manuscript) and has never been documented in macaques in this facility.*

7. Was the fact that animals returning to the breeding colony remained Chagas disease free a surprise given that the parasites seem to transmit freely there?

Reinfection is of course a possibility. However, the incidence of new infections in this colony is variable, but in the range of 1-2% of animals/year (so reinfection risk is relatively low). Also, these cured animals might be expected to be more resistant to (re)infection than naïve animals were (e.g. they have been "vaccinated" by the infection/cure), making it even less likely they would become reinfected. However, this issue of reinfection in previously cured

animals has not been explored in any natural infection system, thus it remains of interest to continue to follow these animals in the future.

8. On page 8 line 20, giving the animals food treats is mentioned. How often and what were those food treats? If ad hoc rather than scheduled this should be discussed either here or in methods. In methods, some of the primate PK work used pumpkin slurry. IS this a standard procedure known not to influence PK?

Drug is incorporated into food treats so the animals will willingly take the drug (and not require daily intubation). This is detailed in the M&M. The food treats were varied in order to maintain animal interest and reduce suspicion (these monkeys are really smart). The standard/most popular is pumpkin balls (hence the use of pumpkin slurry in the PK studies) but peanut butter balls and frozen banana balls are also used as needed.

9. Some more stats would be helpful on the cardiac issues identified since 2 treated (out of 19) versus one untreated (out of 3) would indicate that the incidence in treated is either less or certainly of no significant difference than in untreated, but it would be good to show the workings.

The study was not designed (and is not powered to evaluate) cardiac issues – these are just noted for completeness of information (as are weight change, etc). The compounds are being evaluated for the ability to impact infection, not to modify clinical disease. A study focusing on the latter would require a much different design.

10. The Discussion, page 9 line 32 I would change "the" apparent target to "an" apparent target as polypharmacology of the compound can't be ruled out at this point. Same on line 5 of page 10.

Good point, done.

Reviewer #4:

The manuscript entitled "Discovery of AN15368, An Orally Active Benzoxaborole as a Potential Treatment for Chagas Disease" by Padilla et al. describes the discovery and preclinical characterization of the drug candidate AN15368 belonging to the class of benzoxaborole for the treatment of Chagas disease. Chagas disease remains a significant Global Health R&D priority, as the available therapies, to manage and prevent the disease, are broadly viewed as inadequate. Novel and better therapeutics to treat Chagas are urgently needed. Given the scarcity of Chagas drug discovery efforts, the results reported here are thus significant advances in the field, in particular as it relates to the use of a

Nonhuman Primate model of Chagas disease which has not been previously used to support drug discovery.

The manuscript provides a detailed description of the medicinal chemistry optimization from the previously reported lead AN4169 (reference 4) to the ultimate candidate AN15368. The authors provide a compelling dataset supporting the successful multi-parameters optimization of this chemical series, notwithstanding a more in depth description of this work would be valuable, although more appropriate for a more specialized medicinal chemistry journal.

We agree, this is in the works.

Following on the previously published finding that *T. cruzi* carboxypeptidase can cleave and activate ester containing pro-drug (reference 21), the authors provide strong genetic evidence that the serine carboxypeptidase TcCBP is required for the conversion of AN15368 to its active carboxylic acid metabolite AN14667.

The authors provide evidence that AN15368 is potently active against *T. cruzi* in a variety of in vitro and in vivo assays. Although, the lack of positive controls, such as the standard of care (e.g. SOC, Benznidazole or Nifurtimox) makes it difficult for the reader to appreciate the significance of the results achieved in some of those experiments (e.g. Fig 1B-D and Fig 3D). The preclinical safety assessment is perfunctory and the in vitro safety data should be provided as supplemental information. The safety study in rodent is only for seven days and the authors do not establish the potential target organs for this drug class as no histopathology is reported for any of the doses tested. The reader cannot relate the provided exposure data at the NOAEL (AUC =30,000 ng*hr/mL reported page 6 line 36, is this total of free AUC?) to the potential exposure anticipated to be required for efficacy in humans. Finally, there are no safety margins estimate to help the reader appreciate the relative safety of this novel chemical entity. Note that based on monkey exposure data provided, these margins appears to be potentially small (e.g. < 3).

We disagree that side-by-side comparisons to benznidazole and nifurtimox are useful for the screening assays, although we have evaluated both drugs in many of these same types of assays. We have added additional references in the study referring to these comparative data in these assays. With respect to safety assessment, additional preclinical work is of course needed prior to human clinical trials; these studies have been significantly delayed by Pfizer's acquisition of Anacor but are on course to be completed. For the studies already completed, the observed impact in rats were observed microscopically in liver (dose dependent), kidney (high doses only) and spleen (at high dose only). AUC at the high dose of 600 mg/kg was

256,000 ng*hr/mL. AUC at the NOAEL dose was 30,000 ng*hr/mL. All the AUC reported in this paper are the total of the protein bound and the bound fraction of the drug. NHPs and humans have approximately 2-fold higher free fraction compared to rats. Also, it is notable that as there were no accumulation of the drug seen in rats during the 7-day dosing phase, it is unlikely that the AUC will be very much different during longer term dosing. As noted in the supporting online material, the dose of 30 mg/kg in NHPs was ~48-fold higher dose than predicted efficacious dose in NHPs (based on allometric scaling from the mouse efficacious dose of 2.5 mg/kg). In AUC comparison this exposure at 30 mg/kg in NHPs was ~5.5-fold higher than the exposure at a NOAEL dose in rats, when plasma protein binding difference is taken into account [$F_{unbound}(\text{rat}) = 10\%$, $F_{unbound}(\text{cyno}) = 21\%$].

The most interesting and novel part of this manuscript reports the efficacy of AN15368 in a Non-Human Primates (NHP) Chagas disease model. Because these animals are naturally and chronically infected with a broad range of *T. cruzi* genetic types (or DTU) and display some of the pathology associated with chronic Chagas disease, this model could potentially be more accurately reflecting the clinical situation of chronic indeterminate Chagas patients. The authors decided on a 60-day treatment regimen without a strong rationale other than to say that this is the standard treatment regimen for patients treated with benznidazole. The dose of 30 mg/kg used is also poorly justified. Why did the authors choose the target of > 1 ng/ml for 20-24 hrs? What is the PK/PD model/hypothesis that they used to make these decisions? Why did they believe that achieving this concentration, regardless of whether this is protein-bound or free plasma concentration, would be efficacious? Notwithstanding, the authors provide compelling evidence that AN15368 treatment for 60 days at 30 mg/kg is well tolerated and the compound is able to clear the infection in almost all animals in both blood and tissues. Moreover, monitoring of antibodies suggest that all animals seroconvert after treatment, as anti-*T. cruzi* antibodies decline over a 42 months period after the end of treatment. This is very important because seroconversion is a critical endpoint for clinical trials in Chagas patients. Unfortunately, this study has a number of shortcomings that do not allow the reader to draw any conclusions with respect to the potential advantages of AN15368 to treat the key target population of chronic indeterminate Chagas patients.

First, the authors should provide insights into the efficacy of the standard of care Benznidazole in this NHP model.

There are no published results on use of benznidazole in NHPs although we have conducted such a study in cyno macaques with ~30% cure rate, similar to results in humans. We have added a line to the Discussion indicating this unpublished result.

Second, a more thorough assessment of the histopathology in chronically *T. cruzi* infected NHP would help assess whether one could expect a reversal of some of the pathologies associated with chronic infection in Chagas patients. Only three untreated animals were analyzed, and one animal cured spontaneously without treatment, this small number of control animals do not allow a robust analysis of the interesting heart histopathology findings reported in data S1 table. Do the animals treated with AN15368 resolve most of these symptoms upon parasite clearance and seroconversion? The authors with this current study do not address this important question, and it would be a very significance advance to the field if a larger negative control cohort had been used.

The focus of this drug discovery effort was on the identification of a compound that would provide parasitological cure. The study is not powered to address any alterations in clinical disease that resulted from the treatment, nor would one expect such reversal of disease. Such a study would require 3-4-times the number of animals used here and a potential follow-up period of decades.

In the final section of this manuscript, the authors show that, like other previously described oxaboroles developed for the treatment of Sleeping Sickness (refs 7, 19 and 20), AN15368 likely inhibits the *T. cruzi* Cleavage and Polyadenylation Specificity Factor (CPSF3) enzyme involved in the regulation of mRNA processing and maturation. The authors provide genetic evidence that overexpression of CPSF3 results in significant but moderate level of drug resistance to AN15368. In support of this hypothesis, the authors provide data showing a specific decrease in *T. cruzi* mRNA expression upon AN15368 treatment. Collectively these data are consistent with previous reports but do not provide direct and more definitive evidence that CPSF3 is the actual direct target of AN15368. It is notably regrettable that the authors do not provide more insights into potential drug resistance liability through drug pressure selection that could reveal clinically relevant drug resistance mutations in TcCPSF3 and/or TcCBP.

*We have added a new panel e to Fig 5 showing that mutating the apparent drug binding site in *T. cruzi* CPSF3 provides resistance to AN15368. Generation of stable, increased resistance to AN15368 or analogues by slowly increasing drug pressure has been very difficult. These year-long experiments are on-going.*

Overall, this manuscript provides a good overview of the discovery of a novel agent for the treatment of Chagas disease and provide the first description of a NHP model for Chagas disease. This is an important report and addressing several of the issues highlighted above would make this manuscript more attractive to the general readership of Nature Microbiology.

Minor comments:

1. The section page 2 lines 21-25 is a very long sentence and could be rewritten.

Done

2. Adding short headings for sections would help the reader following the paper.

Done

3. Page 3. Lines 1-2 suggest that metabolic stability tracked with clogD but most compounds reported in Table S2 have low clearance regardless of clogD values.

Changed, this reference should have been to the Table S1 benzyl esters

4. Page 4 line 2 : Change "Several" to 4 (as only 4 of the 10cpds have 10 ug*hr/ml AUC)

We prefer to start the sentence with "Several" rather than a number, and the meaning is not different.

5. Page 4 lines 6-12: hasn't this data been published before?

No, we have published some data on AN4169 (reference 5) but nothing on any of these compounds.

6. Page 5 line 26 – "greatest" does not seem necessary here.

Changed to "particular"

7. Fig.3D. it would be great to see the images as supplemental data

Done. Representative images have been included as Supplementary figure 3.

8. Page 6: line 35-36: Rat PK: FigS4. i.v PK is missing, also provide the Cmax, AUC, F in tabular form.

Done

9. Page 6: line 43-44: Monkey PK: Fig. S5. i.v PK missing, also provide the Cmax, AUC, F in tabular form

Has been added

10. Page 9 lines 16-23: it is intriguing to see some of the analogs active against extracellular but not intracellular amastigotes. Could this be due to permeability issues? Can the authors provide measurement of permeability of these compounds? Alternatively, is it possible that AN15368 is not cleaved efficiently by human host (but not in other non-human species) and that could be the reason for good activity against Tc and no other trypanosoma species (Page 10 lines 8-15)

As indicated in the text, the activity on extracellular but not intracellular amastigotes suggests limited permeability to these compounds into mammalian cells. Alternatively, these could be rapidly cleaved by host cells and thus never reaching the intracellular amastigotes. We have not tried to distinguish between these possibilities. As indicated In Fig 2A and the text around that, We hypothesized that the esters might be pro-drugs to the carboxylic acid and were cleaved within either the host cell or by a parasite enzyme. The high sensitivity of extracellular amastigotes to AN14353, but not to the carboxylic acid AN14667 (Fig. 2A) virtually eliminated the requirement for a host peptidase in pro-drug activation.

11. Supplementary info: Most references appear to be cited incorrectly or perhaps the references for the supplemental information are missing.

12. In vitro assays section; please check references (6 describes Sleeping sickness; 24 describes Crypto)

References fixed.

13. In vivo assays section: Cure assay: check the references (ref 25 is for apicomplexan parasites and ref 23 describes toxoplasma). More details on the in vivo experimental protocols would be helpful. (e.g. beginning of the treatment, number of days treatment was given, immunosuppression regimen, imaging schedule).

References have been checked and corrected; M&M checked to make sure all information is present

14. Page 3: CPSF 3 cloning: Check references: ref 9 describes AAT; RNA seq check references. 28, 29 20 31– ref 28 describes spontaneous Chagas;

15. Page 4: 32 ref is missing in references list.

16. Page 5, 3rd para – PK reference is missing.

All fixed

17. Protocol on DTU typing and growth inhibition testing are not provided – kindly include in supplemental information.

done

18. Monkey PK: Fig. S5. i.v PK is missing and please provide the C_{max}, AUC, F in tabular form.

Same as point 9 above – addressed.

Decision Letter, first revision:

Our ref: NMICROBIOL-21123140A

3rd June 2022

Dear Rick,

Thank you for submitting your revised manuscript "Discovery of AN15368, An Orally Active Benzoxaborole as a Treatment for Chagas Disease" (NMICROBIOL-21123140A) and please accept our apologies for the time it has taken us to contact you with a decision on your manuscript. It has now been seen by the original referees and their comments are below. The reviewers find that the paper has improved in revision, and therefore we'll be happy in principle to publish it in Nature Microbiology, pending some revisions to satisfy the referees' final requests and to comply with our editorial and formatting guidelines.

Thank you again for your interest in Nature Microbiology Please do not hesitate to contact me if you have any questions.

Sincerely,
[Redacted]

Reviewer #1 (Remarks to the Author):

none [mentioned to the Editor that "I am content that the authors have addressed my criticisms/comments well."]

Reviewer #2 (Remarks to the Author):

The discovery of a potential new compound for the treatment of Chagas diseases is exciting and high impact and could be a game changer for the field. The paper should clearly be published but the description of the statistical analysis is still not sufficient and needs to be revised before publication. The authors have addressed some of my original concerns, and I appreciate the improved discussion and the addition of amastigote data added to Figure 2. With respect to statistics, every figure legend must be revisited to provide the following types of details: 1) meaning of error bars - number of replicates, and whether they report SEM or SD (for example figure 2a-c) 2) Stars are used to depict

p-value cutoffs in several figures but they are only defined in Figure 1. A statement is made in the methods that T-tests were used, but it should be made clear that this analysis covers all of the figures where p-values are provided. Please also check that the number of mice/rats has been defined in each figure legend where animals are used.

I've provided only examples for some of the figures. But all of the figures need to be revised/checked to ensure they provide appropriate description of the statistical analysis, number of replicates, etc.

It would be helpful to define KO on first use.

Reviewer #3 (Remarks to the Author):

I consider the authors to have satisfactorily addressed the questions posed after the first round review. The work reports a very exciting finding with regards to a new drug for Chagas disease and is comprehensive in nature.

Reviewer #4 (Remarks to the Author):

Thanks to the authors for resubmitting a revised version of the manuscript. The authors addressed all minor points and provided a reasonable set of rationale for most of the major points. They recognize the importance of the histopathology findings reported and provide a compelling argument for not being able to provide more insights into the cardio-pathology observed in the NHP model as they chose to focus on parasitological cure. Notwithstanding, two areas of concerns remain unaddressed.

1_ The authors disagree with the need for head-to head comparison with approved drugs for most of the assays reported. We would have to agree to disagree that the inclusion of positive controls in their experiments are useful, especially when it relates to NHP studies. It is generally good scientific practice to include positive controls when available, and this would provide the general reader with a very useful frame of reference for the important discovery reported here. Although, the authors provide references to published results and unpublished NHP in vivo data with benznidazole, the authors prefer to not include this data in this manuscript.

2_ The authors do not provide further justification of the dose used in the NHP study. One of the very important contribution of this manuscript is the description of this new NHP model and it would be useful to see a more robust analysis of the pharmacological dose response relationship and the rationale for the dose used in the various animal models. The description in the M&M is quite perfunctory and the allometric scaling method used is far from being optimal as it uses a simple conversion of doses based on body weight and body surface and it does not consider the species pharmacokinetic differences. I could not see any changes in the manuscript and/or comments in the point-by-point responses to the reviewers addressing the pharmacology questions raised.

Decision Letter, final checks:

Our ref: NMICROBIOL-21123140A

13th June 2022

Dear Rick,

Thank you for your patience as we've prepared the guidelines for final submission of your Nature Microbiology manuscript, "Discovery of AN15368, An Orally Active Benzoxaborole as a Treatment for Chagas Disease" (NMICROBIOL-21123140A). Please carefully follow the step-by-step instructions provided in the attached file, and add a response in each row of the table to indicate the changes that you have made. Please also check and comment on any additional marked-up edits we have proposed within the text. Ensuring that each point is addressed will help to ensure that your revised manuscript can be swiftly handed over to our production team.

In recognition of the time and expertise our reviewers provide to Nature Microbiology's editorial process, we would like to formally acknowledge their contribution to the external peer review of your manuscript entitled "Discovery of AN15368, An Orally Active Benzoxaborole as a Treatment for Chagas Disease". For those reviewers who give their assent, we will be publishing their names alongside the published article.

Nature Microbiology offers a Transparent Peer Review option for new original research manuscripts submitted after December 1st, 2019. As part of this initiative, we encourage our authors to support increased transparency into the peer review process by agreeing to have the reviewer comments, author rebuttal letters, and editorial decision letters published as a Supplementary item. When you submit your final files please clearly state in your cover letter whether or not you would like to participate in this initiative. Please note that failure to state your preference will result in delays in accepting your manuscript for publication.

Cover suggestions

As you prepare your final files we encourage you to consider whether you have any images or illustrations that may be appropriate for use on the cover of Nature Microbiology.

Covers should be both aesthetically appealing and scientifically relevant, and should be supplied at the

best quality available. Due to the prominence of these images, we do not generally select images featuring faces, children, text, graphs, schematic drawings, or collages on our covers.

Nature Microbiology has now transitioned to a unified Rights Collection system which will allow our Author Services team to quickly and easily collect the rights and permissions required to publish your work. Approximately 10 days after your paper is formally accepted, you will receive an email in providing you with a link to complete the grant of rights. If your paper is eligible for Open Access, our Author Services team will also be in touch regarding any additional information that may be required to arrange payment for your article.

Please note that *Nature Microbiology* is a Transformative Journal (TJ). Authors may publish their research with us through the traditional subscription access route or make their paper immediately open access through payment of an article-processing charge (APC). Authors will not be required to make a final decision about access to their article until it has been accepted. [Find out more about Transformative Journals](https://www.springernature.com/gp/open-research/transformative-journals)

Authors may need to take specific actions to achieve [compliance with funder and institutional open access mandates](https://www.springernature.com/gp/open-research/funding/policy-compliance-faqs). If your research is supported by a funder that requires immediate open access (e.g. according to [Plan S principles](https://www.springernature.com/gp/open-research/plan-s-compliance)) then you should select the gold OA route, and we will direct you to the compliant route where possible. For authors selecting the subscription publication route, the journal's standard licensing terms will need to be accepted, including [self-archiving policies](https://www.nature.com/nature-portfolio/editorial-policies/self-archiving-and-license-to-publish). Those licensing terms will supersede any other terms that the author or any third party may assert apply to any version of the manuscript.

[Redacted]

Best regards,

[Redacted]

Reviewer #1:

None

Reviewer #2:

Remarks to the Author:

The discovery of a potential new compound for the treatment of Chagas diseases is exciting and high impact and could be a game changer for the field. The paper should clearly be published but the description of the statistical analysis is still not sufficient and needs to be revised before publication. The authors have addressed some of my original concerns, and I appreciate the improved discussion and the addition of amastigote data added to Figure 2. With respect to statistics, every figure legend must be revisited to provide the following types of details: 1) meaning of error bars - number of replicates, and whether they report SEM or SD (for example figure 2a-c) 2) Stars are used to depict p-value cutoffs in several figures but they are only defined in Figure 1. A statement is made in the methods that T-tests were used, but it should be made clear that this analysis covers all of the figures where p-values are provided. Please also check that the number of mice/rats has been defined in each figure legend where animals are used.

I've provided only examples for some of the figures. But all of the figures need to be revised/checked to ensure they provide appropriate description of the statistical analysis, number of replicates, etc.

It would be helpful to define KO on first use.

Reviewer #3:

Remarks to the Author:

I consider the authors to have satisfactorily addressed the questions posed after the first round review. The work reports a very exciting finding with regards to a new drug for Chagas disease and is comprehensive in nature.

Reviewer #4:

Remarks to the Author:

Thanks to the authors for resubmitting a revised version of the manuscript. The authors addressed all minor points and provided a reasonable set of rationale for most of the major points. They recognize the importance of the histopathology findings reported and provide a compelling argument for not being able to provide more insights into the cardio-pathology observed in the NHP model as they

chose to focus on parasitological cure. Notwithstanding, two areas of concerns remain unaddressed.

1_ The authors disagree with the need for head-to-head comparison with approved drugs for most of the assays reported. We would have to agree to disagree that the inclusion of positive controls in their experiments are useful, especially when it relates to NHP studies. It is generally good scientific practice to include positive controls when available, and this would provide the general reader with a very useful frame of reference for the important discovery reported here. Although, the authors provide references to published results and unpublished NHP in vivo data with benznidazole, the authors prefer to not include this data in this manuscript.

2_ The authors do not provide further justification of the dose used in the NHP study. One of the very important contributions of this manuscript is the description of this new NHP model and it would be useful to see a more robust analysis of the pharmacological dose response relationship and the rationale for the dose used in the various animal models. The description in the M&M is quite perfunctory and the allometric scaling method used is far from being optimal as it uses a simple conversion of doses based on body weight and body surface and it does not consider the species pharmacokinetic differences. I could not see any changes in the manuscript and/or comments in the point-by-point responses to the reviewers addressing the pharmacology questions raised.

Final Decision Letter:

Dear Dr Tarleton,

I am pleased to accept your Article "Discovery of an Orally Active Benzoxaborole Prodrug Effective in the Treatment of Chagas Disease in Non-human Primates" for publication in Nature Microbiology on behalf of {redacted}, who is currently travelling. Thank you for having chosen to submit your work to us and many congratulations.

Over the next few weeks, your paper will be copyedited to ensure that it conforms to Nature Microbiology style.

After the grant of rights is completed, you will receive a link to your electronic proof via email with a request to make any corrections within 48 hours. If, when you receive your proof, you cannot meet this deadline, please inform us at rjsproduction@springernature.com immediately. You will not receive your proofs until the publishing agreement has been received through our system.

Acceptance of your manuscript is conditional on all authors' agreement with our publication policies (see <https://www.nature.com/nmicrobiol/editorial-policies>). In particular your manuscript must not be

published elsewhere and there must be no announcement of the work to any media outlet until the publication date (the day on which it is uploaded onto our website).

Please note that *Nature Microbiology* is a Transformative Journal (TJ). Authors may publish their research with us through the traditional subscription access route or make their paper immediately open access through payment of an article-processing charge (APC). Authors will not be required to make a final decision about access to their article until it has been accepted. [Find out more about Transformative Journals](https://www.springernature.com/gp/open-research/transformative-journals)

Authors may need to take specific actions to achieve [compliance with funder and institutional open access mandates](https://www.springernature.com/gp/open-research/funding/policy-compliance-faqs). If your research is supported by a funder that requires immediate open access (e.g. according to [Plan S principles](https://www.springernature.com/gp/open-research/plan-s-compliance)) then you should select the gold OA route, and we will direct you to the compliant route where possible. For authors selecting the subscription publication route, the journal's standard licensing terms will need to be accepted, including [self-archiving policies](https://www.nature.com/nature-portfolio/editorial-policies/self-archiving-and-license-to-publish). Those licensing terms will supersede any other terms that the author or any third party may assert apply to any version of the manuscript.

As soon as your article is published, you will receive an automated email with your shareable link